# *ELiT*, Multifunctional Web-Software for Feature Extraction from 3D LiDAR Point Clouds

**Sergiy Kostrikov [1,2,*]**, **Rostyslav Pudlo [2]**, **Dmytro Bubnov [2]** and **Vladimir Vasiliev [2]**

[1] Department of Human Geography and Regional Studies, School of Geology, Geography, Recreation and Tourism, V.N. Karazin Kharkiv National University, 61022 Kharkiv, Ukraine

[2] EOS Data Analytics, 61002 Kharkiv, Ukraine; rostyslav.pudlo@eosda.com (R.P.); dmitriy.bubnov@eosda.com (D.B.); vladimir.vasiliev@eosda.com (V.V.)

[*] Correspondence: sergiy.kostrikov@eosda.com or sergiy.kostrikov@karazin.ua; Tel.: +38-066-753-5325

**Abstract:** Our research presents a complete R&D cycle—from the urban terrain generation and feature extraction by raw LiDAR data processing, through visualizing a huge number of urban features, and till applied thematic use cases based on these features extracted and modeled. Firstly, the paper focuses on the original contribution to algorithmic solutions concerning the fully automated extraction of building models with the urban terrain generation. Topography modeling and extraction of buildings, as two key constituents of the robust algorithmic pipeline, have been examined. The architectural scheme of the multifunctional software family—EOS LIDAR Tool (*ELiT*) has been presented with characteristics of its key functionalities and examples of a user interface. Both desktop, and web server software, as well as a cloud-based application, *ELiT* Geoportal (EGP), as an entity for online geospatial services, have been elaborated on the base of the approach presented. Further emphasis on the web-visualization with Cesium 3D Tiles has demonstrated the original algorithm for efficient feature visualizing though the EGP locations. Summarizing presentation of two thematic use-cases has finalized this research, demonstrating those applied tasks, which can be efficiently resolved with the workflow presented. A necessity of a conclusive workflow elaboration for use cases, which would be based on the actual semantics, has been emphasized.

**Keywords:** LiDAR; building model; DEM-G/AFE classifying algorithm; *ELiT* software; Geoportal; 3D Tiles; thematic use cases; 3D LiDAR point cloud

---

## 1. Introduction: Initialization of 3D City Models in Urban Studies through Lidar Data Processing

### 1.1. Common Issues

Seeking innovative solutions in urban studies primarily implies to implement novel approaches, technologies, and techniques applied in the information systems and remote sensing domains. Thus, the urban remote sensing and the relevant data processing and modeling for urban studies, can hardly be overvalued. The current global urban monitoring and mapping with usage of the Earth Observing means is the dominant approach that guarantees to avoid the shortage of reliable spatial data of any scale and resolution for urban areas of any size and locations [1–3].

These days 3D City Models are employed in applications in a wide variety of subject domains in both human geography, and urban studies, as well as in environmental science and landscape architecture [4–6]. In addition to the mentioned major sciences, we can easily indicate the relevant usage of these models in applied fields of physical geography, municipal planning, civil engineering, agricultural design, forestry, environmental policy, ecology, regional economics, demography, sociology, and in other domains. These modelled entities can demonstrate overall performance

in up to thirty use cases, that cover more than one hundred applications [7]. Innovative Web-GIS tools intended to create three-dimensional city models and analyze a distribution of their spatial sets afterwards can contribute with this technique to the areas like urban disaster/catastrophe management, health care, telecommunications, facility management, classification of processes in different hierarchical urban geosystems, and social/human /environmental monitoring for the smart city concept implementation [8,9]. In our opinion, such an approach may substantially contribute to outlining adequate ontological frameworks that would combine the urban information and the relevant knowledge due to a standardized modelled presentation of actual environment for a given city in its various scopes, while, the techniques produced on the base of this approach do assist in making city efficient models [10,11].

A 3D City Model possesses the commonly accepted status as a geospatial entity of the certain standards (e.g., Open Geospatial Consortium ones) [1,4–6,8,11–15]. Thus, it can only be a result of a geographical information system (GIS) processing pipeline. A GIS has had a historically close relation with urban studies and as well as with many other affiliated subject fields, because both these subject areas were to the certain extent implemented as quantitative trends in the general geography. Nonetheless, despite understandable GIS-advantages in the contemporary urban studies and continued rapid increase of the relevant social/environmental information the results are stored digitally in numerous different software packages, which use heterogeneous data types and formats. These formats are often defined exclusively according to local necessities and temporary needs of a given project. Moreover, a recurring usage of this data is often impossible exactly, because of missing information about a way in which data are stored, their representation and structure, finalized quality, the date to which the information refers, the scale employed, and due to several other factors. All mentioned circumstances normally lead to widely isolated geodatabases used in urban studies, if only the relevant standardized procedures are not involved. An example of such key procedure can be an introduction of the CityGML standard [16–18].

We have mentioned above the key importance of the urban remote sensing (URS) as of an information source for generation of three-dimensional models possessing the topology, geometry and texture of urban features. The URS techniques can be spaceborne, airborne, and terrestrial platforms, that employ multispectral and hyperspectral tools, as well as radar ones [19]. Light detection and ranging (LiDAR/lidar) technology is a number of methods measuring ranges or distances due to time differences between transmitting/receiving laser impulses [3]. The data obtained with lidar remote sensing as 3D point clouds are normally dense and of high accuracy, and quite long before this technique was stated to be the most preferable one for the efficient urban feature extraction [20–23]. With drastically increasing requests for highly accurate 3D city models and corresponding digital elevation models (DEMs) and due to enlarging availability of airborne lidar (ALS)/terrestrial (mobile) lidar (MLS)/drone lidar (UAV-LS) data, three-dimensional urban features, and buildings, first of all, have become the most prominent entities of urban environment modeled with lidar pipeline processing [3,24–27].

Obviously, 3D city models as representations of a 3D solid volume of urban environment can be obtained from alternative sources [7], but exactly LiDAR data have been recognized as the most preferable ones according to relatively low cost, evident universality, and high precision [28–37].

3D LiDAR point clouds accumulated after the surveys provided over urban areas are the origin for applying the automated feature extraction (AFE) technique, when the points from different reflective objects are isolated from each other, in other words—filtered. Two main classes of points resulted after applying filtering procedure as the ground class, and the non-ground one, while the latter becomes a source for building model extraction [25,28,35,38–43]. The ultimate AFE result can be accepted as a set of building models, which is possible to place within the general frameworks of 3D city models by choosing both an appropriate modeling approach, and a way of their presentation in the variety of virtual city models [44,45].

Approaches and methods that extract urban features and buildings in any alternative ways are of great interest, as it is very promising for various applications of 3D city models. The key range of these applications can, for instance, consist of urban planning, population estimation, urban disaster management, energy sector, planning of infrastructural networks, outlining different smart city projects, and solutions with 3D city models due to visibility analysis in urban environment. Automated feature extraction from the point clouds collected over urban areas is an extremely challenging task for its developers, surveyors, and other researchers, since it means 3D automatic mapping with a corresponding 3D scene generation, which should represent a space of the highest complexity. What is more, exactly LiDAR surveying technique has become for approximately three latest decades an efficient alternative data source for automated building detection, extraction and reconstruction with quite different methodologies and algorithms [3,46,47]. Thus, the building and other man-made feature extraction from point clouds together with the relevant digital elevation model (DEM) generation is one of the most challenging research and development aims for proliferating urban studies as well as for support of making decisions for the urban environment by means of digitalization and information networks.

The presentation of the models of buildings within the frameworks of 3D city models in various zooms of a 3D Scene implies the necessity of powerful visualizing tools' usage, that should be feasible for specification of a web-geoinformation platform, and be able for streaming huge geospatial 3D datasets, basing on the WebGL technology. Cesium 3D Tiles seems to be the most preferable 3D Web-GIS platform, that suggests a virtual globe for visualizing the massive geospatial dynamic data volumes [17,48,49]. Several open-source solutions have been already provided for converting various models into 3D Tiles, what allows substantial optimization for modeled results streaming and rendering [50].

Within the frameworks of our research and software development activity we accept a common definition of the urban remote sensing with LiDAR as that technology, which can be used to acquire the primary data for further processing, and generate the derivative information about the topographic surface, an urban vegetation belt, and various features of the human infrastructure (buildings, bridges, roads, powerlines, etc.) in a selected area of interest (AOI) [3,9,11,19–21,23,26,51]. Such understanding of this technology can be associated with either block, or district scope, as well as with a whole city one. We have already presented in several papers published some key characteristics of our multifunctional approach to airborne\terrestrial\UAV lidar data processing with purposes of fully AFE and DEM generation [8,9,52–54].

## 1.2. Automated Extraction of Building Models and DEM Generation

Traditionally the urban remote sensing has dealt with spectral imageries and with the photogrammetric point clouds. Since high positioning accuracy became available for lidar hardware almost three decades ago, and due to drastically lowering this hardware cost, that has been seen for recent year, the relevant surveying technique is gradually becoming preferable for modeling according to the urban monitoring necessities [21,23,25,55].

Significant advantages for such solution may be defined by that circumstance, according to which LiDAR sensors of various hardware platforms can deliver point datasets with huge ranges of point densities (e.g., varying from only a few up to several thousand points). Even with the lowest values of these densities range, it may be possible to extract urban features, their exact boundaries, and topographic characteristics. Those models can be created, which correctly simulate building facades, and roof structures. A number of relevant techniques for surveyed lidar data processing have been developed according to necessities of urban topography generation and 3D building reconstruction [56–68].

In some of our previous works we have already generally classified the existing AFE-methods on the base of a primary data source [52–54]. The first way implies a treatment of the high-resolution airborne imageries with supplementary including DEMs into an algorithmic work-flow [11]. Although

with such method some substantial results have been received, some references can be found, according to which the "exclusively aerial imagery" solution may be accepted as that one, which does not always perform well enough in the urban areas with dense housing, and detected skewed errors are primarily caused by landscape gaps, shadows, and contrasts in different urban configurations [60,61]. Thus, the AFE procedures based exclusively on the first approach may not be fully effective in the robust applications. The second way straightforwardly involves lidar data and techniques, and it can obtain the definitely improved AFE-output, if compared to the imagery-only methods [26,51]. Methods and procedures applied within the third way in a common case combine both aerial imageries, and various types of lidar surveys (ALS/MLS/UAV-LS) in order to use combined information from all data sources [62,63].

One more structuring of the automated feature extraction methods can be provided on the base of the building rooftop detection, segmentation and reconstruction [58]. Thus, the model-driven AFE for building roofs is provided, when either a predefined catalogue of roof templates-primitives does exist [11,46,63] (e.g., it may be the CityGML rooftop catalogue [16]), or some formalized description for of the features to be extracted is provided, e.g., a determination of the parametrized shapes in a 3D space with 3D Hough transform methods [64]. Another interesting solution within the model-driven paradigm is referred to in [65], where 3D models of prototypical roofs are based CityGML LOD2 templates, what has given an opportunity to perform through sparse point clouds [65].

The data-driven AFE procedures are often named as generic approaches [30], which often mean the feature extraction directly from digital elevation/digital surface models (DEM/DSM) [41,65]. The data-driven methodology may also be defined as a variety of polyhedral-delineating methods, since the feature model produced may consist either from few, or from many polyhedrons [40,58,66]. Since this text is not a review paper, we can mention with only few additional references, that in our opinion, just the data-driven methods, and those ones, which combine model-driven and data-driven approaches, do provide the majority of effective solutions in automated feature extraction from LiDAR surveyed data [58,67–69].

It directly proceeds from the primary basics of remote sensing with Lidar [3,19], as well as this point of view has been proved in a separate reference concerning the definite advantages of DEM producing with Lidar results instead of photogrammetry [70], that a generation of a digital elevation/digital surface model is an inalienable part of a whole AFE-pipeline, even if the filtering technique is considered as a dominant component of the relevant workflow [71]. It seems to be understandable, that in the most cases a Lidar DEM is resulted not in a set of urban features, but in an aggregate of those ones, which are purely topographic [72]. Nonetheless, in this case modeled topographic surface can be used as a scene basic layer for man-made discrete features extracted.

*1.3. Some AFE and DEM Creation Problematic Issues*

While we presented in a brief manner earlier the original algorithmic flowchart of building detection, extraction, and reconstruction within the high polyhedral modeling frameworks of building simulation by ALS data processing, we emphasized that ground may have its original characteristics that refute ground/non-ground feature distinguishing depending on the certain location and given terrain conditions [52,54]. Normally the following features often refute filtering/classifying algorithms: low vegetation belt; low walls, which are along sidewalks; bridges; nonstandard buildings; hill cut-off edges; complex mixed land covering, especially—by man-made features; areas combined with low and high-relief terrains. All these features cause the lack of reliable accuracy estimation upon LiDAR data processing for AFE. If the feature extraction belongs to "coarse" research techniques by default and should be applied, first of all, to large geospatial datasets [73], the finalized requirements to the derivative data output have to be sooner overrated, than underrated. Introducing the two-branched DEM generation—AFE classifying algorithm in our text pursues this goal besides all the other: to meet enhanced requirements to the algorithmic output accuracy.

While finalizing the literature review, it would be reasonable also to examine some other problematic issues addressed in our research in the AFE/DEM creation workflow, if accomplished through LiDAR data processing. The separation while processing ground points from non-ground ones is mostly titled as a filtering procedure [74]. For almost all lidar pipelines, ground detection and filtering are a united mandatory step to determine which ones of the LiDAR returns are from the bare ground surface, and which ones are from non-ground surfaces, which understandably belong to the discrete features. Thus, a ground surface we can define in a similar way as a continuous feature. Distinguishing the ground from the non-ground can be normally a task of significant difficulty in regions with the high topographic variability. Moreover, it is necessary to take into account that a reliable DEM can only be constructed, if non-ground points are removed before the interpolation provision through an initial grid [71].

The urban topography is the physical basis for the various urban feature allocation. Proceeding from following references that highlight just this subject area [56,70,75], it is necessary to mention, that the earth surface topography possesses two key premises useful for its reconstruction on the base of ground points presented in a 3D LiDAR point cloud: (1) this surface is both continuous, and piecewise-smooth one. It is presented throughout a whole scanned area and over it, with each its point georeferenced (X, Y—two flat georeferencing coordinates, and the vertical coordinate Z, normally associated with surface | feature height); (2) there are no other scanned points at some significant distance under the surface, despite some noise points and outliers caused by mirroring from other surfaces, vertical wells, rock walls, etc. Even if there are some abnormal ground areas (e.g. vertical and overhanging ones), they are bound continuously with the dominant, more horizontal surface, where "into the underground" direction has been defined uniquely.

According to several references the key characteristics of the issued ground surfaces can be classified into four such sets based on general properties of topographic surfaces [76–78]. All four key parameters imply crucial significance exactly for an urban terrain generation, when non-ground features usually are densely presented in the neighborhood. Thus, a key premise is to distinguish ground points from those ones of neighboring urban features, and this premise has to be completed in that segment of a whole AFE workflow, which filters prescribed point classes out. Four mentioned parameters are as follows: (1) the lowest absolute heights (altitudes); (2) the steepness of topographic surface; (3) the altitude difference of ground surface; and (4) the homogeneity of topographic surface. All these characteristics are applied withing the framework of our two-branched algorithmic solution suggested in this text.

The finalized problematic issue relates to the existing wide range of building sizes may cause problems for some filters employed in the classifying procedures [76]. Topographic filters based on the running window algorithms sometimes have problematic situations removing large or small constructive features. This situation may appear, because the classifying filters distinguish points proceeding from a comparative analysis between the value measured and the value estimated within a certain outlined neighborhood. If a large construction is completely placed within a running window, the cloud points appearing in the middle of a corresponding footprint may not be taken as a part of this construction, because there may be no significant difference between two types of values—the measured and the estimated ones.

The main goal of this paper is to outline the research contribution to the generation of building models and their further application by presenting the relevant complete R&D cycle—from raw 3D lidar point cloud processing for AFE purposes, and up to thematic use cases' implementation on the web-portal. First of all, presented research implies consideration of the high polyhedral modeling technique for the fully automated AFE with a DEM generation for urban areas. Thus, two key constituents of the whole workflow are examined with further making an emphasis on some aspects of the urban feature web-visualization and brief reviewing two thematic use cases. The original geoinformation web- and cloud-based software platforms developed for the listed purposes are described in our paper too.

## 2. Approach and Methods: Urban Topography and Building Model Extraction from Airborne 3D LiDAR Point Clouds

*2.1. High Polyhedral Modeling and Two-Branched DEM Generation/AFE Algorithmic Solution*

We have already outlined in our earlier publications two of our somewhat alternative AFE solutions as the high polyhedral modeling (HPM) of buildings and the low polyhedral one (LPM) [52–54]. These two modeling techniques are key constituents of the authors' lidar data processing work-flow, but in general they may identify two substantial mainstreams in various existing AFE approaches. Although the case is that feature extraction techniques earlier applied, as a rule, were not targeted to set up themselves primarily according to existing urban configurations, while our the HPM methods presented briefly in [52,53], and the LPM ones introduced in [53,54] in details do within the multifunctional R&D approach resulted in the relevant web-software elaborated. The multifunctionality of our AFE-technique implies not only its applicability to different urban configurations (e.g., high-rise buildings of city central parts, and low-rise buildings of suburbs and rural areas), but also various functional software tools, what is described below.

A text introduced in this paper chapter concerns our own original both conceptual, and algorithmic solutions just within the high polyhedral modeling frameworks. It implies the production of building models, which surfaces consist of a big number of polyhedrons, and therefore the relevant modeled results can be accepted as heavyweight models. It means, an HPM-building model may be generated from up to more, than hundred thousand of points. Once we attempted to prove, that for the HPM frameworks the point cloud classification procedure is the dominant one, and it is not directly associated with clustering, while for the LPM operations the point cloud segmentation through clustering is the key one [54]. The LPM building models generated can be composed of not so many facets, and the number of points prescribed for one only model generation is limited by a number of approximately five thousand. A reasonable number may be obtained by adaptive thinning techniques at the cost of details.

Upon the implementation of our original ground filtering/point classifying algorithm two alternative algorithmic approaches have been considered, evaluated, and updated, which elements in one way or another are presented in the following references [40,56,70,75–81]: (1) The choice of the thinned network of the antecedent ground points. The set of relatively large parcels, within which we can easily select at least one ground point, is considered. (2) The parcels of the non-ground points have to be filtered out from the network of the points densely located. The first approach may hardly classify properly those points that are located around the common edge of the data spatial extent. It requires the more, the better those overlapping areas of classified parcels, which have to be processed independently. If there is in the spatial extent selected within a large area with no points at all, this large parcel may be mistakenly classified as the ground. The second approach requires the very accurate differentiation the connected ground parcels from the non-ground features located on the topographic surface. It is normally caused by the smooth surface transition from the ground to the non-ground features (front entrance to a building; contiguity of roofs with the topography; embankments, the walls that are not vertical ones; etc.). All abnormal topographic features mentioned above should also be taken into account.

The first algorithmic approach has been chosen after detailed analysis as a more reliable one. Its formalized view within our general Lidar point cloud classifying approach is presented on the following flow-chart (Figure 1). This flow-chart mirrors our original two-branched urban DEM generation (DEM-G)/automated feature extraction algorithmic solution, which consists of the Non-Ground (AFE) algorithmic (filtering/classifying) branch (the left one in Figure 1) and the Ground (DEM-G) algorithmic branch (the right one). According to the consequent steps in an overall filtering/classifying workflow the Ground branch has to be described the first, notwithstanding we have placed it to the right, taking into account by way of example the specificity of kindred the TEXAS (terrain extraction and segmentation) algorithmic workflow [82].

## 2.2. Ground (DEM-G) Classifying Algorithmic Branch

According to the flowchart of our original algorithm shown in Figure 1 the step-by-step description of its Ground branch may be like follows with respect to a descending numeration of the relevant flow-chart blocks for this branch.

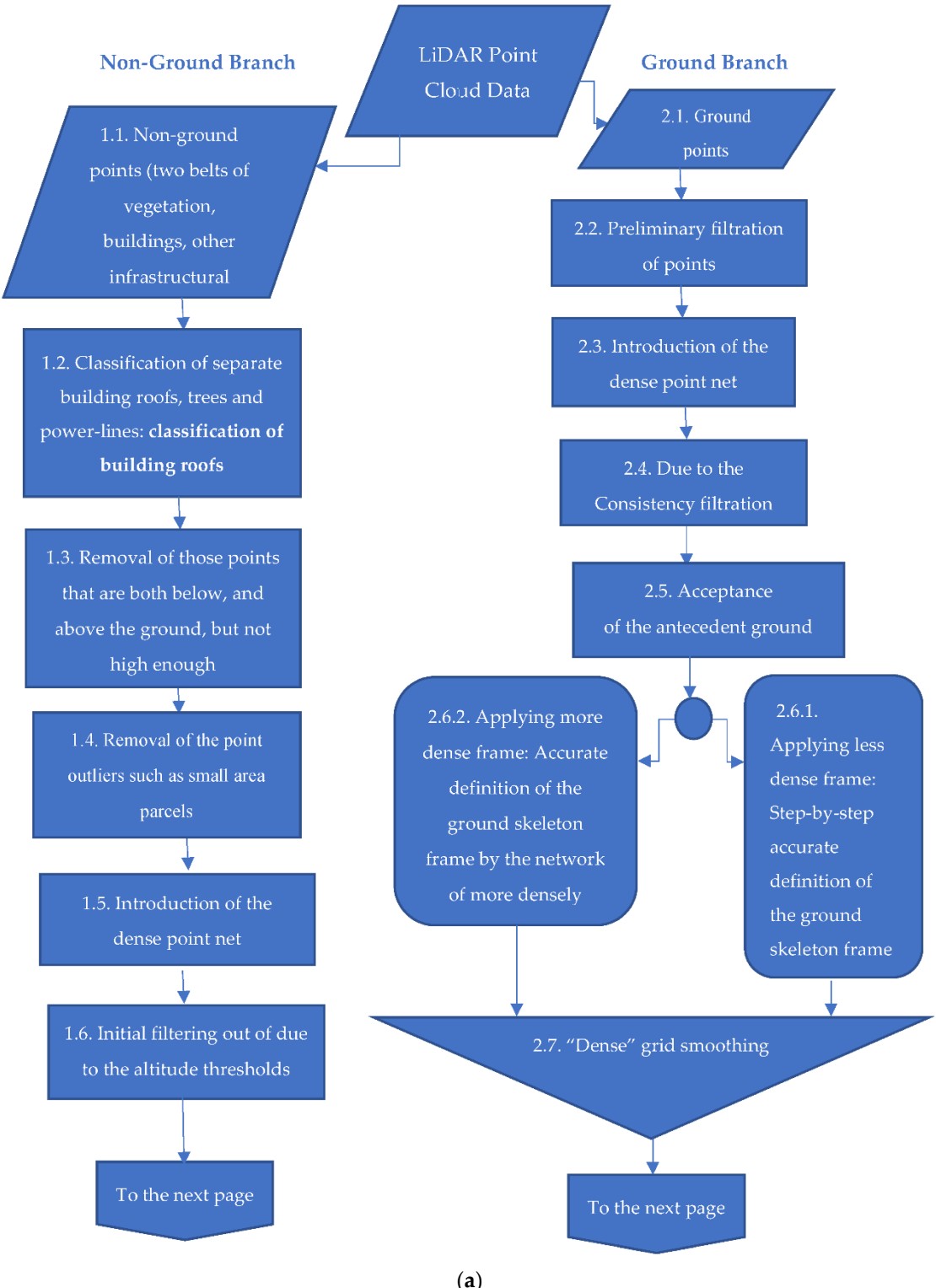

(**a**)

**Figure 1.** *Cont.*

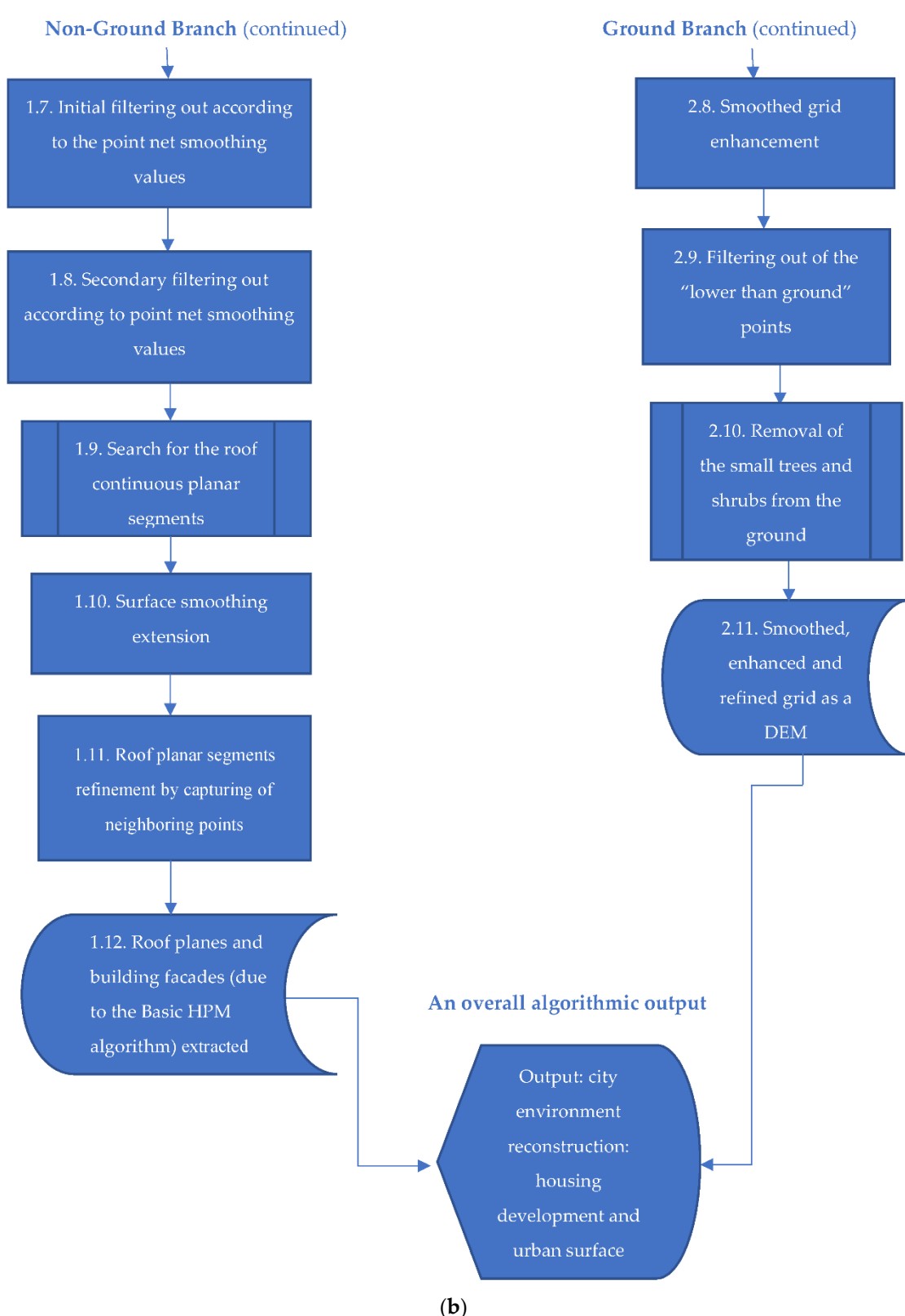

**Figure 1.** (**a**) The upper part of the flowchart due to the two-branched digital elevation models generation (DEM-G)/automated feature extraction (AFE) algorithmic solution within the high polyhedral modeling (HPM) frameworks. Only airborne lidar (ALS) data are processed according to this flowchart. (**b**) The lower part of the flowchart due to the two-branched DEM-G/AFE algorithmic solution within the HPM frameworks. Only ALS data are processed according to this flowchart.

- Ground points (block 2.1 of the Ground branch—Figure 1):
  It implies an input of that LiDAR point cloud for processing, which contains some ground points by default;
- Preliminary filtration of points (block 2.2 of the Ground branch):
  The ground classifying algorithms are normally used for analysis of the point mutual location, thus either their semi-coincidence, or complete coincidence would negatively impact classification results. Therefore, point semi-coincidence (or duplication) may distort a point array drastically, and one of two points duplicated have to be marked as a "noise" point, while a set of such points—"doubles" may form together an outlier.
- Introduction of the dense net of the points (block 2.3):
  It should be taken by default, that all points lying on the topographic surface are lower, than any other points related to those features, for which the topography is the base. Thus, it is necessary for ground classifying to keep the lowest points only, and these points have to be bound to relatively small ground parcels. Normally, even upon the smallest lidar point density (1–2 points per square meter) we have to select the lowest point within a parcel of 2 m × 2 m, and using the sliding window method.
- Due to the Consistency (Point Density) Filtration (block 2.4):
  All the points selected by now within both algorithmic branches (refer to Figure 1a,b) belong to reactively smooth, un-transparent features (either to ground, or to building roofs). They have almost uniform distribution of their density along the whole data extent. That allows to build a triangulated irregular 2-D network (TIN) throughout all these points. All those edges, that are too long, or that have some pitch according to the normal to the topographic surface should be removed from this TIN. This procedure determines and excludes those small abnormal topographic parcels (topographic sinks and gaps, shaft wells, sharp small peaks), which break the network smoothness with respect to this network unit height distribution;
- Acceptance of the antecedent (reference) ground points (block 2.5):
  Those points left after block 2.4 completion are named as the reference points. These points should not expose sharp sinks, thus each of them should initiate a starting node of a newly smoothed surface construction, and this surface can be large enough. The antecedent ground points can be chosen by default in the following way. The common data extent should be partitioned for the parcels large enough to include ground points, but each surface area raised above neighboring parcels ("a hill") and delineated by surface breaks, should completely include such parcel, which, "is large enough". All neighboring parcels mentioned are located for a half of each size shift one relatively to another; thus, they intersect one another for a half of its area. The key issue is that there should not be parcels crossed by the common area spatial extent edge upon a whole surface partition. Such parcels should be ignored, since the point presence within these areas cannot be taken by default. Normally, the lowest point has to be selected within the all point parcels obtained upon former algorithmic steps. This point is selected after the removal of 0.3% of lower points that considered as random noises/fluctuations. The parcel size of 30–50 m along each its edge allows to classify efficiently even a sharply crossed topographic area, upon a condition that there are no big buildings within it. In case, when large building constructions are present (approximately with the roof size of 50 m × 50 m and more), the parcel size should be enlarged up to a size of the biggest building so that to avoid an evident spatial contradiction. We have to take into account that those parcels, which are either with broken terrain, or even with crossed topographic surface ones, as well as those ones with buildings present and of small area, may not be classified as ground points. The total length of each whole data extent edge for any side of a parcel as a partitioning result should at least trice exceeds a length of the side of this parcel, where antecedent ground points are located. Otherwise, the points selected may be localized within a small area only, and this area cannot be the base for a whole local topography construction, while

these points cannot be the reference points.

The next algorithmic block (block 2.6) is the only one which branches out for "two sub-blocks".

- Step-by-step Accurate Definition of the Ground Skeleton Frame (block 2.6.1, the right sub-block of the sixth block—Figure 1):

    Thus, a TIN is constructed through the reference points, and its facets characterize the topographic slope on some local parcel. For each point selected upon the third algorithmic block the nearest facet should be found, while **z**-coordinate of this facet is not taken into account. The distance–height between the point and the facet should be measured, as well as an elevation of this point above the highest point of this facet. This distance–height is negative if the point is below the facet. The height of the highest point, which belongs to this facet, is measured too. The lowest point among all those ones, which belong to this facet, is selected. All selected points, which become the reference ones, should not be higher above the facet surface more than for a certain value, and should not be more distant from the facet, than for a certain value. Upon these measuring procedures both the point distance to the facet, and its height above the surface of the facet should be accepted with equal weighting coefficient of 0.5. Such acceptance allows smoothing drastic slopes of the reference surface obtained from the reference points (refer to block 2.5), on which these slopes tend to appear along this surface edges. The points selected are consequently added to the reference TIN, and the next iteration is completed until then, when all points are being successfully gathered. The total number of iterations should not exceed some threshold value defined. Those points that are too close to the points already added should be ignored. Thus, we could avoid the topographic breaks in an approximating TIN. In this way, the ground surface built through the almost lowest ground points is defined more precisely by the lowest points newly added step by step, while the approximating procedure can be completed through the smooth topographic elevations, but it cannot be—through the topographic breaks and walls.

- Accurate Definition of the Ground Skeleton Frame by the Network of More Densely Located Points (block 2.6.2, the left sub-block of block 2.6):

    Following parameters for "2.2–2.6 ground" classifying algorithmic blocks should be entered for corresponding processing into the relevant Macro Library dialog (MLD) with further implementation in our desktop application *ELiT*Core—Figure 2.

    After the completed topographic skeleton-frame of the ground surface has been built as a continuous one on the point network of low density, this frame can be made to become even denser by applying the same method of a sliding window of 0.5 m × 0.5 m for making a network denser. The same algorithm, that has been applied in the third algorithmic block, is employed once again, if the point density value is acceptable for provision of this procedure. Since our accepted size of the sliding window as 2 m × 2 m contains 16 cells of 0.5 m × 0.5 m, a total number of algorithmic iterations is not too big in this case.

- "Dense" grid smoothing (block 2.7):

    Despite expectations that continuous topographic skeleton-frame might be built precisely and accurately up till now, it may include some abnormal topographic deviations (which are, as a rule, of lower altitude, than necessary), because they may not be filtered out in the fourth block. A procedure of "dense" grid smoothing for that topographic skeleton-frame, which has been already obtained as a compacted ("highly dense") one, allows to eliminate these deviations in the same manner, as it has been done in block 2.3.

- Smoothed grid enhancement (block 2.8 of the Ground branch—Figure 1):

    After the smoothed grid has been obtained with the corresponding topographic skeleton-frame, this frame should be enhanced by the neighboring points. These points are both those ones from the network of the lowest points of that sliding window 0.5 m × 0.5 m cell mentioned, and all other points, which lie beyond this window edge, if their density is satisfactory.

    Nonetheless, before immediate enhancement of an obtained grid, it may be necessary either to build through the topographic frame mentioned above some another interpolated grid, or apply

exactly this direct TIN of the skeleton-frame for other features' classification, if this classification requires some additional information. It may be a case, when precise classification requires some knowledge about feature allocation in relation to the earth surface. If it is not provided, the further processing may appear to be a cumbersome one.

- Filtering out of the "lower than ground" points (block 2.9):
  By default, all points that are lower, than those ones classified as "the ground", should be defined as the "lower noise" (topographic sinks, shaft wells, other negative mirroring of laser sensing). Customized input parameters for "2.7–2.9 ground" classifying algorithmic blocks should be set up and entered for corresponding processing into the relevant MLD in the *ELiTCore* software, just as it has been done for blocks 2.2–2.6 (Figure 2).

- Removal of the small trees and shrubs from the ground (block 2.10):
  This one before last algorithmic block of the Ground branch finally refines the topographic surface modeled through blocks 1–9. Thus, the next derivative results are conclusively obtained in the last block of this algorithmic brunch.

- Smoothed, enhanced, and refined grid as a DEM (block 2.11):
  In this way an "urban DEM" (urban terrain) is created, which we understand as a synonym of a digital terrain model, which represents the bare earth ground with uniformly spaced **z**-values within any "urban area".

## 2.3. Building Extraction (BE) Classifying Algorithmic Branch

The Non-Ground branch, the left brunch of a two-branched algorithmic flowchart (Figure 1) can be also defined as the building extraction (BE) classifying branch due to a subject of its algorithmic content. The two-branched algorithm presented exposes key details of a point cloud classification as a fundamental basic of the high polyhedral modeling of buildings. Both branches, the ground and non-ground ones, belong to one logically united algorithmic workflow, and they are separated in the text exclusively for the better perception by a reader. That united workflow gives an opportunity to provide combined datasets of building models placed on a DEM of different scales as a final processed result obtained in a fully automated mode, while according to well-known references those CityGML models, that contain both terrain and buildings can hardly be overvalued [83].

In general, several robust classifying approaches exists. Only for example, one straightforward approach may be like follows. We emphasized in the previous subsection the significance of choosing a proper preprocessing method for raw LiDAR data, while obtaining a grid. If this grid surface is a satisfied one, then the next algorithmic procedure may be related to removal of points of some heights below a certain value (probably, 2–3 m). Normally, it is the ground peaks, some human infrastructural and other features (bridges, roads, and vehicles), low belt vegetation (shrubs, bushes, etc.). Thus, all these points are eliminated from a LiDAR dataset, while those points that left and primarily consist of buildings and high vegetation belt are kept for further processing. Then different segmentation procedures can be applied to these remaining points.

Proceeding from our own processing experience, we can state, that there may be two types of mistakes upon point cloud classification procedures: (1) A feature is not classified to a class it should belong to. In a case of building classification with the high polyhedral modeling it may be caused by some buildings having a roof, that may be not a set of smooth surfaces. For example, there may be some steeples on the roofs, or their decorated relief (e.g., Gothic architecture); (2) The features of other class are mistakenly classified as a given class features. Upon building classification, the high features with smoothed roofs and other smoothed surfaces, that have their facets large enough, can produce the classification mistakes exactly of this, the second type. The actual features may be mistakenly classified as buildings, but they really may be big trucks, a forest segment of smoothed geometric shape and with the dense foliage not allowing for lidar scanning to pass through. Other similar cases are also feasible. Almost all mistakes mentioned can be corrected applying some supplementary improving

criteria of the applied classification. These criteria should be further elaborated, and without these parameters a whole classification workflow described below is the preliminary one only.

In this text, we consider the presented classifying algorithm applied to those points, which belong to building roofs; thus, it is the straightforwardly modified version of our basic feature extraction algorithm [52–54]. The algorithmic blocks presented in the branch described below are shown in Figure 1 above. The building classification follows the ground one; thus, it is understandably based on those points, which are above the ground. We can accept as a template of the topographic ground level, either a level of that TIN surface, which has been built on the reference ground skeleton-frame, or a level of the surface built on an interpolated grid.

According to the flowchart of the two-branched algorithm shown in Figure 1, the step-by-step description of its Non-Ground branch (the first, left, algorithmic brunch) may be like follows with respect to a descending numeration of the flow-chart corresponding blocks for this branch:

- Non-ground Points (two belts of vegetation, buildings, other infrastructural features) (block 1.1 of the Non-Ground branch—Figure 1):
  It implies an input of that Lidar point cloud for processing, which contains at least some non-ground points by default;
- Classification of separate building roofs, trees, and power lines: classification of building roofs (block 1.2 of the Non-Ground branch):
- Removal of those points that are both below, and not high enough above the ground (block 1.3):
  It is understandable, that the first step to remove "abnormal" points would be a step to eliminate points, that are below some height threshold value. The topographic altitude of 1.5–2 m cannot be accepted to be a reliable threshold value, which would indicate the building roofs. These points should be removed.
- Removal of the point outliers such as small area parcels (block 1.4) and initial filtering out of:
  Spatially isolated sets of points ("outliers") that possess relatively small areas (e.g., up to 15 m$^2$) should be eliminated also, since they do not allow to identify definitely a building in comparison, for example, with a big truck.
  At this point of the Non-Ground branch some customized input parameters for "1.2–1.4 non-ground" classifying algorithmic blocks should be set up and be input for processing into the relevant MLD, just as it has been illustrated for blocks 2.2–2.6 above (Figure 2).
- Introduction of the dense point net (block 1.5):
  In the majority of cases the roof surface is not a transparent one. Therefore, most of the roofs of low-rise buildings may belong to the lowest points among all non-ground points delineated within some small parcel selected in a point cloud, while the roofs of high-rise buildings may belong to the highest non-ground points. Those roofs that are too transparent for a LiDAR beam may not be found at all. The selection of each lowest point for a given parcel applying a method of sliding window (a cell) of 0.5 m × 0.5 m would provide the whole algorithmic workflow by this block results with necessary precision. It expedites processing, eliminates excessive number of points, and makes a whole point set more uniform. If a point net is not dense enough, the sliding window matrix may be increased up to 1 m × 1 m, or even to the size of 2 m × 2 m. Unfortunately, the output result reliability goes drastically down in such a case. To apply a cell size which is even bigger, than 2 m × 2 m seems to become completely useless.
- Initial filtering out of due to the altitude thresholds (block 1.6):
  The easiest way to remove main part of those drastic topographic outliers is to enter the certain altitude thresholds: to select one lowest point through the net with cells of twice longer edges, than an initial net has; it means—through the twice thinned net in comparison with initial one; in this way, the lowest point is one from four others that belong to this cell selected; a TIN is being constructed through this thinned point net; those points from an initial net are added to an obtained TIN, which are topographically close to that derivative surface, that would be constructed through this TIN obtained.

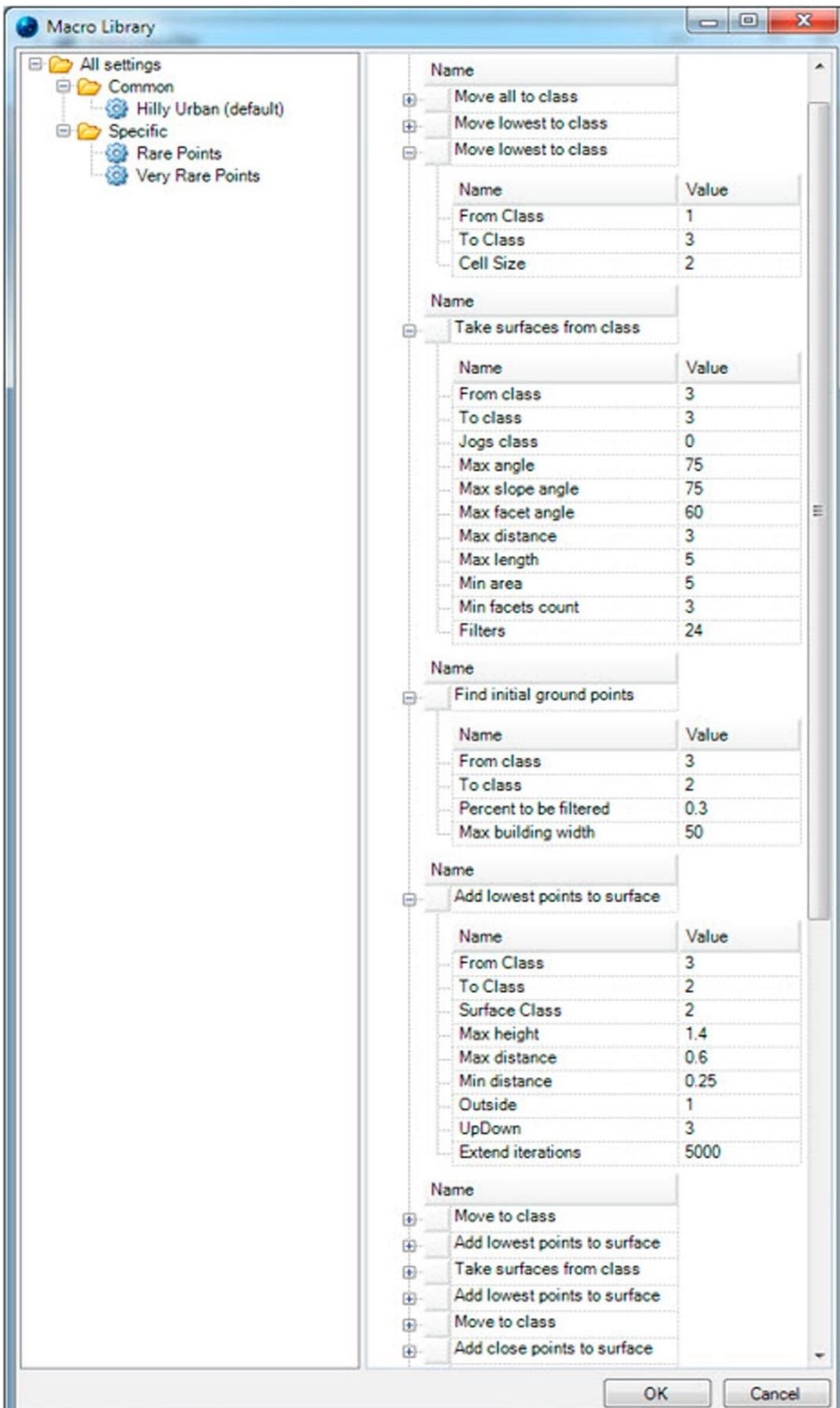

**Figure 2.** The input parameters for the second-sixth classifying blocks of the Ground branch of the two-branched DEM-G/AFE algorithmic solution within the HPM frameworks (Figure 1).

- Initial filtering out according to the point net smoothing values (block 1.7 of the Non-Ground branch—Figure 1):
  A TIN has to be built on the base of points selected in previous algorithmic blocks. All those edges

of triangles that are longer than 1.5 m within X, Y plane, and longer than 4 m in 3D space should be removed from this TIN. After the edge removal the number of TIN facets is substantially decreased. For each of the facets (a "core" facet) left, the angles, each of them makes with neighboring facets, should be considered. Normally the number of neighboring facets changes from 0 to 3. If more than half of derivative angles significantly differ from 0 or from 180 degrees, this "core" facet should be removed. The interactive process of facet removal for satisfactory reliability of the output results should be thricely repeated. Interconnected parcels, which include not fewer, than two facets, and with an area not smaller, than one cell of the initial net (0.25 m$^2$), are selected from that TIN content that has been left after that edge removal. Within the processing in this algorithmic block, the accepted angle between two facets stays within a range from −20 till +20 degree, and from 160 till 200 degrees, either we take in account a negative value of an angle, or not. Bounded parcels with a number of facets fewer than five should be removed.

At this point of the Non-Ground branch some customized input parameters for "1.5–1.7 non-ground" classifying algorithmic blocks should be set up for input due to processing into the relevant MLD as it has been shown earlier for blocks 2.2–2.6 (Figure 2).

- Secondary filtering out according to the point net smoothing values (block 1.8):
  The requirements to the value of the angle between two facets can be made stronger, after initial filtering out of all topographic breaks presented earlier. Filtering out of the previous block is repeated iteratively with allowable angle between the facets of the following range: from −20 to +20 degrees, and from 160 till 200 degrees. Bounded parcels with the number of facets fewer than 5 should be removed.

- Search for the roof continuous planar segments (block 1.9):
  A TIN has to be built on the base of points selected after completing the secondary filtering. All edges with a too long projection on the plane X, Y are removed from this TIN. If the point density is 2 and more per square m, the facet edges, which are longer than 1.5 m, all should be removed. The parcels of interconnected facets with a total area of 15 m$^2$, and which consist of fewer than 40 facets, should be removed. In this way, we eliminate the pints, which occasionally complete a smoothed surface within a small parcel. An increase of the minimal distance between independent parcels and making less strict requirements due to the number of facets within an interconnected parcel may help with processing of some thinned point net of the low point density. From the other side, such solution may substantially increase the probability of the wrong classifying results through those parcels, which occasionally appear to be smoothed ones.

- Smoothing extension of the surfaces (block 1.10):
  We have delineated only some from roof surfaces with cut edges and probably without some invisible roof components upon previous algorithmic blocks (refer to Figure 1). Thus, upon this eleventh algorithmic block we should extent smoothly each isolated roof parcel for account of points in its neighborhood. This procedure consists of several iterations until either the newly generated roof surface reaches the spatial limits defined, or the maximally allowed number of iterations is completed. Thus, we can obtain an enlarged smoothed roof area by combining several roof parcels in one joint as an output result.
  At this point of the Non-Ground branch input parameters for "1.8–1.10 non-ground" classifying algorithmic blocks should be set up for input into the relevant MLD as it has been done for selected sets of blocks above in both branches.

- Roof planar segments' refinement by capturing of neighboring points (block 1.11):
  The finalized procedure of the first algorithmic branch is the enhancement of the roof surfaces by all those points that are on insignificant height from the derivative roof surface. In this way, all those points, earlier removed from building point sets upon previous intentional point thinning are classified as the building ones.

- Roof planes and building facades (due to the Basic HPM algorithm [54]) extracted (block 1.12):
  An overall algorithmic output consists of 3D city models of building features as well as of an

urban surface, which is of high precision (an edge of a grid cell is not more, than 50 cm)—it is represented by a concluding block of the flowchart (Figure 1). Even an attempt of a reconstruction of the "smart urban environment" upon the Smart City concept implementation may be provided on the base of this approach [9].

### 2.4. Multifunctioal Web- and Cloud-Based Software for DEM-G/AFE Purposes

2.4.1. *ELiT*Core Desktop and Web-Based ELiT Server

The *ELiT* (EOS Lidar Tool) is a Web-Based platform titled as *ELiT* Server (a landing page is available on https://eos.com/eos-lidar/), and a cloud-based application that applies to AWS instance resources—*ELIT* Geoportal (EGP). The latter is a type of web portal used to find, access, and process LiDAR geospatial information, both primary, and derivative one. This cloud platform also provides the associated geospatial services (summarizing, display, editing, analysis, etc.), widely using various. Web resources and options (http://ELiT-portal.eos.com/). The precursor software for these two sets of LiDAR data processing/displaying tools is the desktop *ELiT*Core software, to which, although only in terms of embedded MLD, we have already referred to above (Figure 2). This software has even to a certain extent broader functionality, but it can generate only the heavyweight polyhedral models within the HPM framework. The whole AFE-algorithmic pipeline explained above has been firstly implemented just in the *ELiT*Core package. The "building extraction" (BE) functionality based exclusively on the HPM algorithms has been developed for the standalone application for provision of detection, extraction, and reconstruction of heavyweight models according to general workflow of the classifying algorithm introduced in this text and due to the HPM AFE basic algorithm. It results in a topographic grid as a continuous feature generation exactly according to the Ground branch of the classifying algorithmic solution, and in a set of urban features which are the discontinuous objects, which are generated proceeding from the Non-Ground algorithmic branch (Figure 1).

A general architectural scheme of a whole family of the *ELiT* software (a desktop, a web-based server, and a cloud-based Geoportal) is outlined in an illustration below (Figure 3). Implemented web-GIS approach does not contradict to existing samples of implementation of the web-geoinformation tools [84], and similarly to them our solution implies involvement of PostgreSQL/PostGIS as that database management system, which is object oriented.

Thus, the most of computing and processing within the presented in this text architectural software structure is applied to transform raw points into 3D-models. It is necessary to emphasize that these processing is hidden from a user (except some exclusions in the *ELiT*Core desktop), since it is in the pipeline of a higher level. Therefore, a user should only input data and a necessary set of the building modeling characteristics. According the architectural scheme depicted, which is also an operational one, the *ELiT* Server performs transmitting procedures between the algorithmic Core and a web-client, while providing such procedural sets as data management (data uploading, downloading by users, etc.), task management, and interactions between the Core and *ELiT* Geodatabase. Finally, a client provides the user interface and the building model/topographic surface visualization (Figure 3). An open specification for visualizing huge massive volumes of geospatial data Cesium 3D Tiles, already mentioned in the literature review completed due to this text, is employed for this display, but only in our web-based, and cloud-based applications, while the *ELiT*Core software supports routine desktop visualizing options (visuals of building extraction and change detection functionalities in Figure 3).

The purpose of all *ELiT* products is both AFE operations, and topographic grid modeling with further display of urban environment in a chosen area. Processed results are outputted in the different formats (.KML, .gLTF, .DAE, .B3DM), but the main inner format is .OBJ. Modeled urban features are produced with their boundary representation, and a whole picture is a set of buildings with their bounding walls, edges, vertices, and with their topological relations according to non-housing man-made constructions and infrastructural networks. Our urban feature models possess all necessary characteristics of 3D city models, but while many other 3D models seem to be predominantly used for

display, we dare say, that *ELiT* 3D city models can be increasingly employed in a number of domains within a large range of tasks beyond the direct visualization.

**Figure 3.** A general architectural scheme of the EOS LIDAR Tool (*ELiT*) software family: the *ELiT*Core desktop, the *ELiT* web-based Server, and the cloud-based *ELiT* Geoportal.

Four key functionalities of *ELiT* Server are as follows (Figure 3):

- The building extraction (BE) functionality (a sub-page Building Extraction in the Tools page of the Server) provides the high polyhedral modeling according to building detection, extraction, and its reconstruction through that algorithmic solution, which has been presented in detail in the previous section of this text. Finalized building modeling is primarily targeted to high-rise buildings frequently located in city downtowns. The BE-tool of *ELiT* Server provides generation of heavyweight models, consisting of numerous polyhedrons, which is why they may be described as "heavy ones". Finalized visualization of these models is provided by the Cesium 3DTiles library with a certain level of detail (LOD), while a primary Lidar point cloud can be visualized too. As a rule, a BE-model mandatory possesses its spatial, geometric, and semantic attributes. Thus, massive urban environment of a city can be simulated as the heavyweight models with minor details (Figure 4).
- The building extraction rural area tool (BERA)—a sub-page building extraction rural area in the Tools page) completes the low polyhedral modeling introduced in some of our previously published methodological texts [9,53,54]. The BERA functionality accomplishes the hierarchical segmentation of point clouds, and separation of extracted planes with further building reconstruction mainly in rural areas and in urban suburbs. Made "lightweight building models" have substantially fewer facets, than heavyweight ones, and a number of points processed for such model are limited by a number of few thousand only. Accepted number may be reached by adaptive thinning at the cost of some minor details. It is possible to produce even "pitched roofs" of low-rise buildings by the BERA-tool with the LPM-technique (Figure 5).
- The change detection functionality (CD)—a sub-page Change Detection, which is on the Tools *ELiT* page, detects urban alterations of various scales in an AOI selected. Changes in the architectural morphology of a city usually happen through certain spatial extent over some significant period of time, if only it is not any drastic event of environmental or social destruction. The CD functionality indicates locations of changes in georeferenced space and shapes of buildings and infrastructures

as 3D models. Normally two point clouds (the initial, and the second one temporally) are compared. The BE-functionality is the only one that is used to determine the difference between two input point clouds, which is computed as the BE-modeled delta of features, which belongs to each from these two clouds, correspondingly.

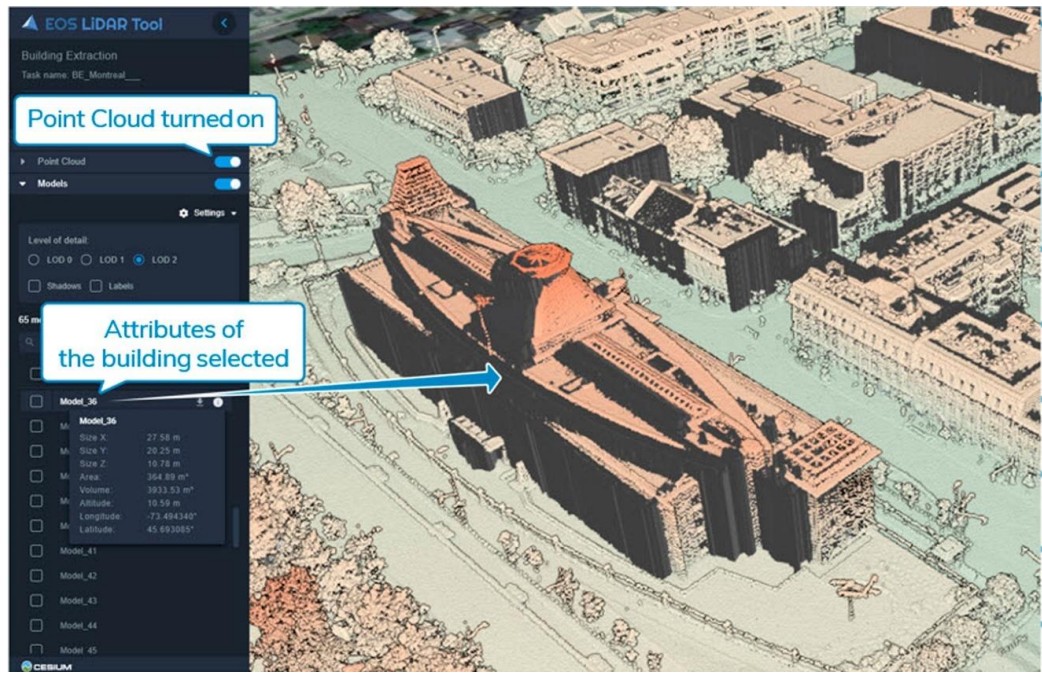

**Figure 4.** Visualization of both a point cloud, and HPM-building models with their attributes for a downtown of Montreal, Quebec, Canada. The visual is presented from the *ELiT* Server web-interface.

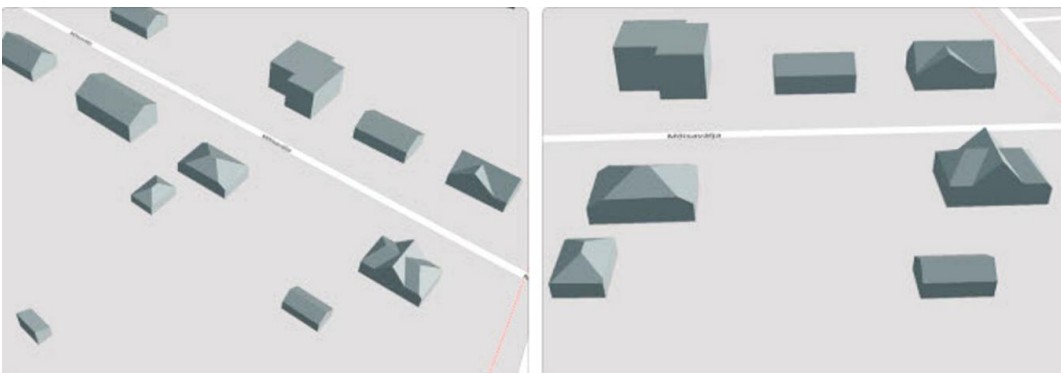

**Figure 5.** Visualization of flat, gable, and pitched roofs that belong to a set of LPM-building models generated for an urban area of Tartu-city, Estonia. The visual is presented from the *ELiT* Server web-interface. A free global map, Open Street Map (OSM) as a global world map, is used as a ground-basis.

- The DEM generation functionality (DEM-G)—a sub-page in the Tools page accomplishes a generation of a grid of a topographic surface by making a DEM/DSM, that mirrors a particular terrain according to the Ground branch of an algorithmic solution introduced in the previous section of this text (Figure 1). By this functionality a user creates a gridded surface from sample data, what is known as the interpolation. Thus, initial irregularly spaced height points are obtained, from which uniformly spaced elevations are interpolated. A digital elevation model we accept as a synonym of a digital terrain simulating the bare earth surface with regularly spaced *z*-values of heights. In this way it is possible to provide topographic modeling for the ground surface of various genetic types, e.g., like that post-glacial topography in the visual below (Figure 6).

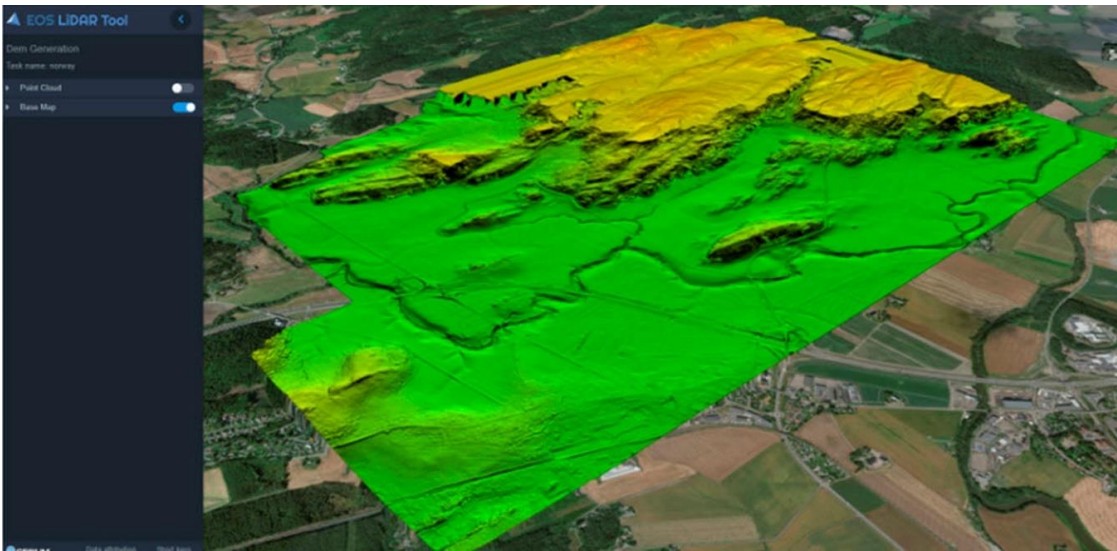

**Figure 6.** Visualization of post-glacial landforms in Norway. The visual is presented from the *ELiT* Server web-interface.

### 2.4.2. *ELiT* Geoportal: Visualizing Urban Features with Cesium 3D Tiles

Once we have already emphasized the efficiency of such robust solution for both the geoinformation web-software, and for its applied services promotion as a Geoportal can be [85]. The *ELiT* Geoportal (EGP) may be outlined as a type of web portal, which is used for finding, accessing, and processing geographic information obtained from Lidar surveying sources. A Geoportal, as a geospatial entity is also targeted to provide the relevant services (initial data storage, derivative data visualization, editing, analysis, etc.) via the Internet. The final goal of any Geoportal as of a geospatial entity is usually to support web-software marketing, which main core is conversion of occasional visitors of this site to its warm leads. What is more, the geoportals are accepted to be the key application of any distributed web-geoinformation system [86].

While developing the EGP as a unique geospatial web-resource intended to store, process and visualize the massive geospatial information content (http://ELiT-portal.eos.com/), we understandably select displaying issues as the key ones, and apply for this purpose to CesiumJS (an open source Java Script library for 3D globe creation, https://cesium.com/), and to 3D Tiles as to that data structure, which makes possible the hierarchical rendering of large datasets with various discrete building models. There are already up to thirty various locations with urban territories of different sizes presented for the time being on the *ELIT* Geoportal. All feature modeled results have been obtained on the base of processing the opensource lidar data, e.g., USGS projects, by our cloud-based software, while employing AWS resources for computing. Key characteristics of the processed 25 locational projects with models according to CityGML LOD 1/LOD2 are presented in Table 1.

Thus, we have to resolve a problematic issue of the 3D Tiles structure creation and its optimization, while visualizing these generated features through huge urban territories. A hierarchical structure of a tileset, as a 3D Tiles key issue, is characterized by JS object notation (JSON)—a JS-based text format of data interchange [50]. A tileset describes an octree of interrelations for all tiles (a spatial index). With respect to the 3D Tiles specification a .glTF file format is used by .B3DM as its payload for transmitting not only 3D geometry of buildings, but also for delivering all that information necessary for the visualization. Thus, it implies ".B3DM = .glTF + attributive information".

For all visualized data, the correct georeferencing is mandatory to possess for a common coordinate system, which is a geographic and Cartesian coordinate system (ECEF—epsg:4978). Even if data are processed not with AWS resources on the Geoportal, but by *ELiT* Server on the localhost, the results of all functionalities—BE, BERA, and CD are displayed by the *ELiT* Viewer on the similar basis. It is necessary to emphasize that only those local coordinate systems can be employed for visualizing,

which can be converted into the ECEF-coordinates by affine transformations. The latter are prescribed by the transforming matrix either in a tileset, or in a .glTF file. A size of .B3DM files may possess key meaning for visualizing efficiency. It directly impacts overloading of a client's RAM. Firstly, we have attempted to enhance the existing Cesium simplified algorithm of 3D Tiles structure building [87], trying to parallelize it, but failed. Then we simply tested this Cesium solution (titling it as (1)-Solution):

1. Start with a bounding box (a root tile) that encompasses all the Geometry. 2. Save relationship between a tile and each Geometry. 3. Add a tile to process queue. 4. Split that bounding box evenly into child quads. 5. For each child quad, repeat until you have n triangles per quad.

It seems like an efficient solution because data are packed very densely into .B3DM files. Nonetheless, such algorithmic solution may cause the existence of following problematic issues:

(1) In a common case, partitioning of tiles with their associated bounding volumes for two different datasets are completely independent procedures with no connections between them. Thus, it is evident that for joining these two already partitioned tilesets we would have either to rearrange completely their tree data structures, or to combine these tilesets with partial overlapping only.

(2) It is necessary to select a proper type of a bounding volume as well as an effective approach for its hierarchical organization.

(3) A necessity to update a tileset (or several of them with new feature units), 3.1; a user updates tilesets in local, or in regional scales, 3.2; it is necessary to update tilesets in a global scale (within a whole Globe), 3.3.

Probable solution for the second problematic issue is as follows. Since we are dealing basically with a certain type of feature models—with models of buildings that correspond to the 3D City GML LOD1/LOD2 standards—it is preferable to select a bounding volume as an "oriented bounding box", a bbox, that precisely fits a 3D Scene spatial georeferenced extent.

The evident solution for meeting challenges of the first and the third problematic issues is in both cases an update of 3D Tiles structure, what only can allow to keep the spatial coherence. An overall update of a whole structure is, as a rule, unacceptable one, because it is too resource consuming. Otherwise, it can be resolved by adding/removing a particular unit in the existing 3D Tiles, or by joining two tilesets. There are up to three different options in this aspect.

The first option: it is necessary to select the biggest possible bbox, which may be of a size up to the size of the Globe. The second one: tiles should be matched in such a way, in which they would possess the same location and size for any initial datasets. It can be obtained by georeferencing to the coordinate origin of the ECEF system. The third option implies the generation of the 3D Tile structure and the content of tilesets (.B3DM files) dynamically. Then, it seems reasonable to implement in our visualizing work-flow some methods from existing and earlier reported techniques [88,89].

At this work-flow point it is necessary to define, if a spatial reference of a bbox to a regular net may help in mentioned above adding/removing a unit to/from existing tilesets, or in joining two tilesets. We create a reference of regular tile partitioning to a net $2^n$, where $n$ is a level of tile partitioning. Referencing to a regular net allows to substitute numerous mathematical operations of division and multiplication for shift operations. It is possible to define a dependency between a tile partitioning level and a LOD (level of detail) of model visualization. The tiles have to be indexed. Each tile has its size ($2^n$) and a cell index according to the coordinate origin. A cell index depends on a partitioning level (on a tile size upon a given level). Each tile index can be defined by a partitioning level and by a cell index (level, $i$, $j$, $k$). Thus, any two tiles of a different tile structure describe the same area, if they have the same tile index. Taking into account both "(1)-Solution" presented above and the technique presented in references [89,90] we have elaborated the original algorithmic approach targeted to visualize and to render efficiently huge tilesets of modeled results presented in Table 1. It has helped to split a calculation into independent parts so that to execute it in parallel mode. Tileset properties for each location imply storing geometric characteristics of buildings (e.g., height and footprint area) in tiles. The visual presents one of the EGP locations (#11) rendered with 3D Tiles (Figure 7).

**Table 1.** Geoportal Key Characteristics of the processed locational projects with the *ELiT* building models according to *CityGML* level of detail (LOD) 1/LOD2 standards.

| # | Project ID in *ELiT* Geodatabase | Project Name of Opensource Lidar Data | Geoportal Location Name | Total Lidar Points | Average Lidar Points Density (PPSM) | *LAS* Files Number | City GML LOD1/LOD2 Number |
|---|---|---|---|---|---|---|---|
| 1 | 6657 | USGS_LPC_MD_PA_SandySupp_2014_LAS_2016 | Baltimore, MD, USA | 12,994,969,727 | 4.47 | 1697 | 51,302 |
| 2 | 6610 | Barcelona | Barcelona, Spain | 67,211,545 | 0.73 | 25 | 10,947 |
| 3 | 6611 | Barcelona (Filtered) | Barcelona, Spain | 319,694,346 | 1.1 | 76 | 112,902 |
| 4 | 5026 | West_Midlands_Birmingham_etc | Birmingham, UK | 941,594,643 | 1.57 | 2088 | 470,349 |
| 5 | 5119 | USGS_LPC_CO_SoPlatteRiver_Lot5_2013_LAS_2015 | Denver, CO, USA | 33,660,346,453 | 5.18 | 6084 | 4939 |
| 6 | 5114 | USGS_LPC_MI_WayneCo_2017_LAS_2018 | Detroit, MI, USA | 6,877,405,127 | 0.51 | 5798 | 86,553 |
| 7 | 5117 | ARRA-MI_4SECounties_2010 | Detroit, MI, USA | 8,438,736,403 | 0.11 | 6318 | 3159 |
| 8 | 5123 | USGS_LPC_MI_31Co_Oakland_2016_LAS_2019 | Detroit, MI, USA | 8.543,602,713 | 0.49 | 8366 | 21,082 |
| 9 | 5122 | USGS_LPC_IN_WT_B9_Lake_2013_LAS_2016 | Gary, IN, USA | 3,809,037,151 | 0.24 | 1304 | 1309 |
| 10 | 5110 | USGS_LPC_IN_MarionCo_2011_LAS_2016 | Indianapolis, IN, USA | 1,852,560,049 | 0.38 | 978 | 38,183 |
| 11 | 5120 | Lyon_ | Lyon, France | 997,975,464 | 5.06 | 286 | 16,846 |
| 12 | 5117 | KS_Area3-NortheastA_2012 | Kansas-City, MO, USA | 5,209,425,053 | 1.13 | 562 | 32,728 |
| 13 | 6609 | Leeds_ | Leeds, UK | 77,338,830 | 7.96 | 42 | 3849 |
| 14 | 5118 | USGS_LPC_CA_LosAngeles_2016_LAS_2018 | Los Angeles, CA, USA | 62,358,249,705 | 0.6 | 26,996 | 3,119,061 |
| 15 | 6612 | Madrid_ | Madrid, Spain | 427,768,563 | 0.79 | 142 | 126,403 |
| 16 | 5023 | USGS_LPC_FL_PalmBeachCo_2016_LAS_2019 | Miami, FL, USA | 7,056,565 | 1.16 | 18,312 | 956 |
| 17 | 5111 | KS_JacksonCo_2006 | Miami, FL, USA | 1,636,615,433 | 1.04 | 3414 | 85,188 |
| 18 | 5025 | FL_MiamiDadeCo_2007 | Miami, FL, USA | 9,094,836,110 | 1.34 | 2442 | 68,188 |
| 19 | 5115 | WI_WisconsinCo_2006 | Milwaukee, WI, USA | 2,502,208,232 | 0.19 | 26,996 | 1755 |
| 20 | 5121 | USGS_LPC_MN_Phase4_Metro_2011_LAS_2016 | Minneapolis, MN, USA | 8,617,642,240 | 10.96 | 3070 | 48,997 |
| 21 | 6656 | Moncton_ | Moncton, NB, Canada | 2,734,944,967 | 15.55 | 176 | 4872 |
| 22 | 5112 | USGS_LPC_LA_UpperDeltaPlain_2017_LAS_2018 | New Orleans, LA, USA | 46,224,930,331 | 4.72 | 9570 | 424,053 |
| 23 | 5109 | KS_JohnsonCo_2006 | Overland Park, KS, USA | 1,206,071,706 | 0.97 | 2694 | 22,028 |
| 24 | 5021 | USGS_LPC_DE_DelawareValley_2015_LAS_2017 | Philadelphia, PA, USA | 38,973,363,102 | 5.16 | 6924 | 5890 |
| 25 | 5113 | USGS_LPC_UT_Wasatch_L4_2013_LAS_2016 | Salt Lake City, UT, USA | 14,496,166,437 | 11.27 | 2750 | 76,835 |

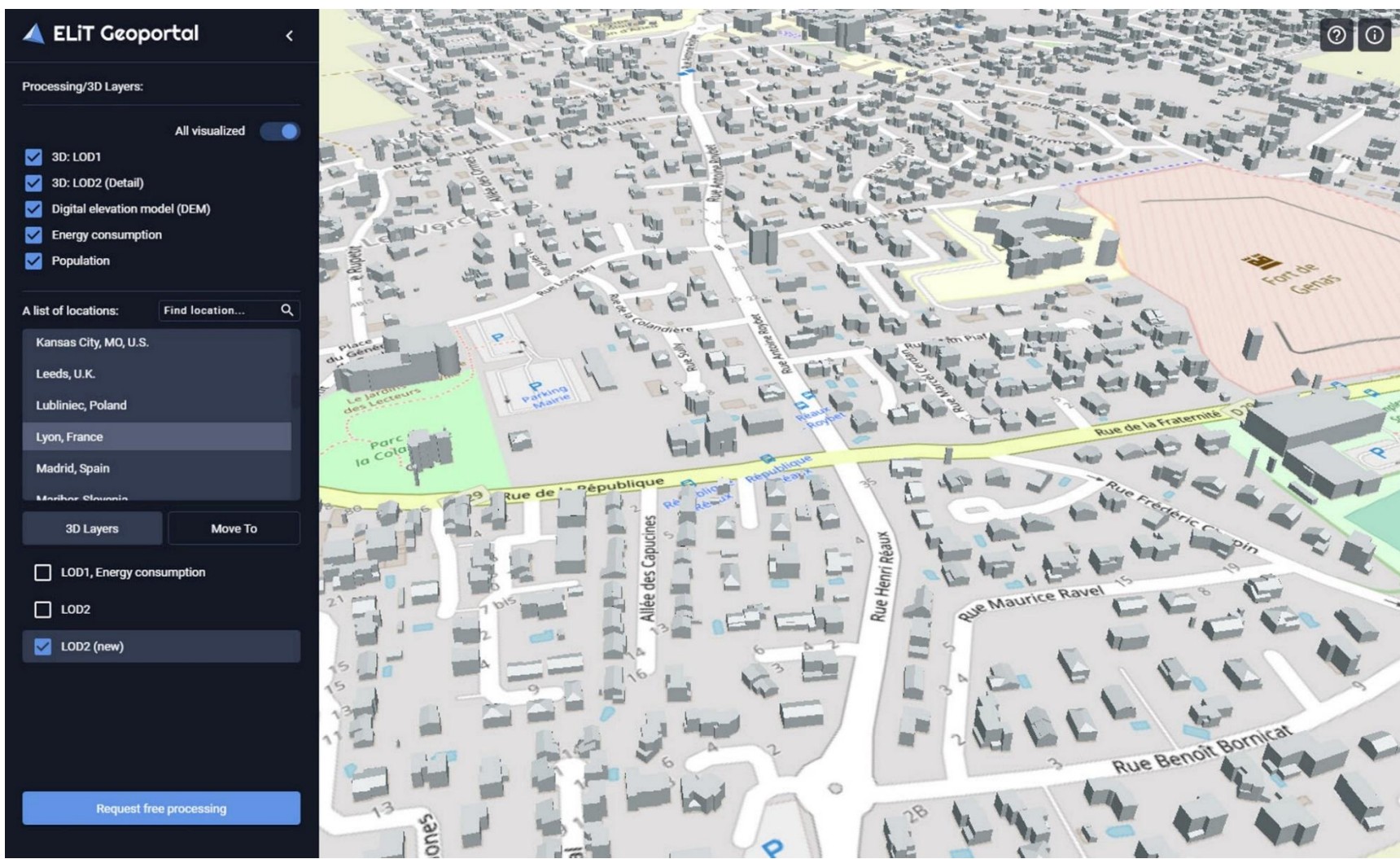

**Figure 7.** A fragment of a whole urban area of Lyon, France presented on the Geoportal as 3D CityGML LOD2 models. OSM is used as a ground-basis.

### 3. Results: Presenting Urban Features and DEM on the Geoportal with Thematic Use Cases

*3.1. Visualising Geoportal Locations*

Our original algorithmic solution for creating the 3D Tiles structure for those urban feature locations maintained on the *ELiT* Geoportal is titled as (2)-Solution (for comparing with "(1)-Solution" we have referred to above). "(2)-Solution" produces a comparable in efficiency with "(1)-Solution" of the 3D Tiles structure, but our solution has been managed to be parallelized. This approach implies that hierarchical data structure (a tile content) is split, what results in a non-uniform overlapping quadtree. The content and the algorithmic sequence of our (2)-Solution and "its advanced option" (there are two options in it—"initial", and "advanced" ones) within it are like follows:

An algorithmic input: sets of models with plain facets. The coordinate system is prescribed for each model as the EPSG code; a list of vertexes, and a set of facets (vertex indexes that define a facet) are prescribed for this model too. Models are stored either in a file system, or in a database.

1.  A bbox has to be defined for each model. For these purposes, each vertex should be transformed into a coordinate system with EPSG: 4979 (ECEF), and a minimal/maximal coordinate value is defined (parallelized computing).
2.  For each bbox its size is defined along each axis and a center of this bounding volume. On the base of a size defined we outline a partitioning level and a position in a tileset. A connection between a tile index and a model identifier is stored (parallelized computing).
3.  A supplementary step. It is necessary to limit a rank of a partitioning level, not fewer than nine, in this way we obtain a tile size (an edge of a square) not fewer than 512 m.
4.  Obtained indexes of tiles and their size values are used for computing a quadtree of tile partitioning (tileset.json).
5.  It is necessary to employ a dependency between indexes of tiles and identifiers of models, so that to compute .B3DM files (parallelized computing for each tile).
6.  The parameters of the first testing package for analysis of "the advanced option" algorithmic results are set for a square edge of 512 m (a size of 1 tile is 0.262144 km$^2$): (1) a number of tiles: 2533; (2) latency time (for a whole data): about one and a half minute; (3) a size of .B3DM files: minimum—3 Kb, (4) maximum—1185 Kb. We can see that this testing package is applicable for an urban are of 664 km$^2$, thus a quite large city can be completely visualized.
7.  The parameters of the second testing package for analysis of the algorithmic results of this option are set for a square edge of 1024 m (a size of 1 tile is 1.0486 km$^2$): (1) a number of tiles: 719; (2) latency time: about a half of a minute; (3) a size of .B3DM files: minimum—3 Kb, (4) maximum—2589 Kb.
8.  It can be seen that this testing package is applicable for an urban area of 753 km$^2$; thus, even a larger urban territory can be visualized with this tile structure.

3D Tiles structures for all 25 locations described with their quantitative properties in Table 1 have been built within both (1)- and (2)-Solutions frameworks. Brief analysis of these tile structures efficiency according to optimized rendering is provided further in this text.

The *ELiT* Geoportal is also enhanced now with a Terrain content for several EGP locations. A corresponding DEM is computed strictly according to algorithmic solutions with the *ELiT* DEM-G functionality, while a Terrain is displayed employing Cesium Terrain Provider (Figure 8).

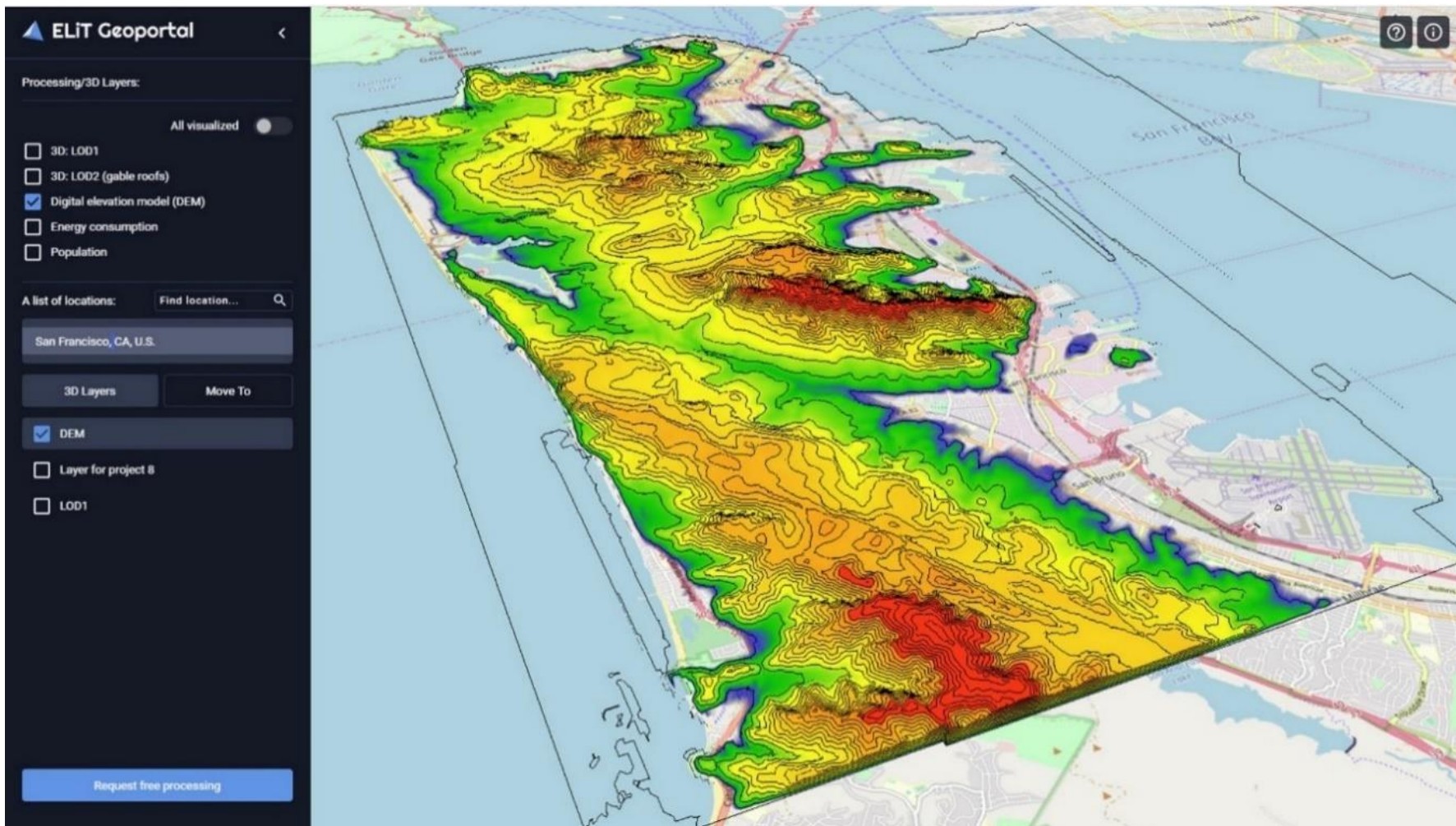

**Figure 8.** An urban area DEM of San Francisco computed on the base of the Ground branch of the two-branched algorithm (Figure 1). It is visualized on the EGP by Cesium Terrain Provider.

*3.2. Thematic Use Cases*

Two completed use case pipelines have been elaborated and implemented on the EGP recently: 1. Population Estimation with Building Geometries—Use Case Population (UCP). 2. Estimation of Energy Consumption by Buildings for Heating and Cooling—Use Case Energy (UCE). Both use cases presented on the Geoportal demonstrate the possibility of their application either in a block, or in a district scope, as well as in a whole city one. Nonetheless, in most projects a particular AOI should be selected. An acceptable size of this text may not allow to introduce both techniques of these use cases implementation in details; thus, we present here only some brief abstract provisions.

3.2.1. Use Case Population

The UCP can hardly be overvalued, because of well-known public scarcity of actual population in various urban configurations in a certain AOI. Moreover, municipalities do not have, as a rule, more or less reliable tools for evaluating population data within a period between two censuses, while this period may last up to ten years in average. Thus, at least approximate population estimations in an AOI are crucially necessary for optimizing preparedness for urgent event in a city, or for enhancement of urban disaster management, what has been evidently confirmed by the course of events in urban areas all over the world upon the contemporary pandemic phenomenon.

Having created the UCP-methodology of population estimation with building geometry, we applied to various econometric and GIS 2D analyses methods, which are mainly based on building footprints and census tracts [90,91]. We also accepted that well-known fact, according to which the urban remote sensing takes its dominant place within a whole relevant pipeline, and LiDAR data processing was highly appreciated exactly for this purpose [92].

The step-by-step building space-metric method (BSMM) of population estimation with building model geometries has been developed just for purposes of this research. If introduced very briefly, the key content of this method consists in the following (what we describe on the example of the EGP location of Boston-city, Massachusetts, USA (Figure 9).

1. Building models of City GML LOD1 standard have been generated from the USGS opensource data project (point clouds of .LAZ format) as sets of .OBJ files and .JSON files. The latter store various relevant metadata. 2. Our own building footprints extracted within the HPM pipeline, which also includes the two-branched classifying algorithm (Figure 1), have been compared with those ones from the OSM resource. Those footprints that matched have been selected as benchmarks for further UCP-processing. Then, a part of benchmarks has been filtered out by a minimal footprint area. 3. All building heights and their volumes have been computed for the BSMM-implementation. 4. Then we have to obtain the most actual census information, if available, and find out, if (1) census tract boundaries either coincide, or intersect the boundaries of these sets of identified benchmarks; (2) census tracts either coincide, or intersect feature parcels obtained from available maps of the urban land-use over the city of Boston. 5. Analyzing information obtained from the thematic maps of urban land use: selecting class of residential buildings versus few classes of non-residential ones. 6. Numbers of storeys have been processed for all benchmark footprints as a quotient of a building height and an average storey height in Boston (2.5–3.5 m). Characteristic values of storey heights have been taken for each census tract from the relevant attributive information of the OSM-source and from some other available municipal sources. Computed numbers of storeys have been compared with actual values, if the latter were available, and have been corrected if necessary. 7. Generating in an opensource GIS (QGIS) a point layer of centroids for all benchmark footprints associated with .OBJ files obtained on Step 1. Bounding with these point features all semantics, both initial, and derivative attributive information. 8. Within the frameworks of the BSMM-introduction a summarized volume of residential buildings in a given census track is computed. Thus, on Steps 6–8 we obtain a layer of census tracts with supplementary attributive information of building volumes (BV) for each of them. This approach is titled as BSMM-BV. 9. Within the frameworks of the BSMM-implementation a summarized number of building storeys (BS) in residential buildings for a given census track is

computed. 10. Applying the BSMM-BV method for computing an approximate number of residents in each residential building. 11. Applying the BSMM-BS method for computing a number of residents in the same buildings. 12. Comparing between themselves results obtained on Steps 9 and 10, removing extreme numbers (too many residents in a small house, only few of them in a large departmental building), comparing results of both steps with at least some actual population numbers if available through census tracts. 13. Choosing appropriate results and adding it to the attributive information of City GML LOD1 models in the Scene (Figure 9).

It seems reasonable to summaries those initial data necessary for the UCP-implementation:

- LAZ point clouds;
- OSM-footprints converted into a single ESRI .SHP;
- three vector layers (.SHP) with 2010 Census data—a layer of census tracts, a layer of sets of city blocks (a group of blocks), a layer of single blocks [93];
- the thematic vector layer of land use for the state of Massachusetts [94].

In addition to the EGP location of Boston, taking into account the available semantic information, the population estimation with building geometries use case has been provided for the city of Chicago, USA, and for Munster, Germany. That we can see in the Geoportal interface (Figure 9), where A list of locations has been filtered by the Population checkbox turned on.

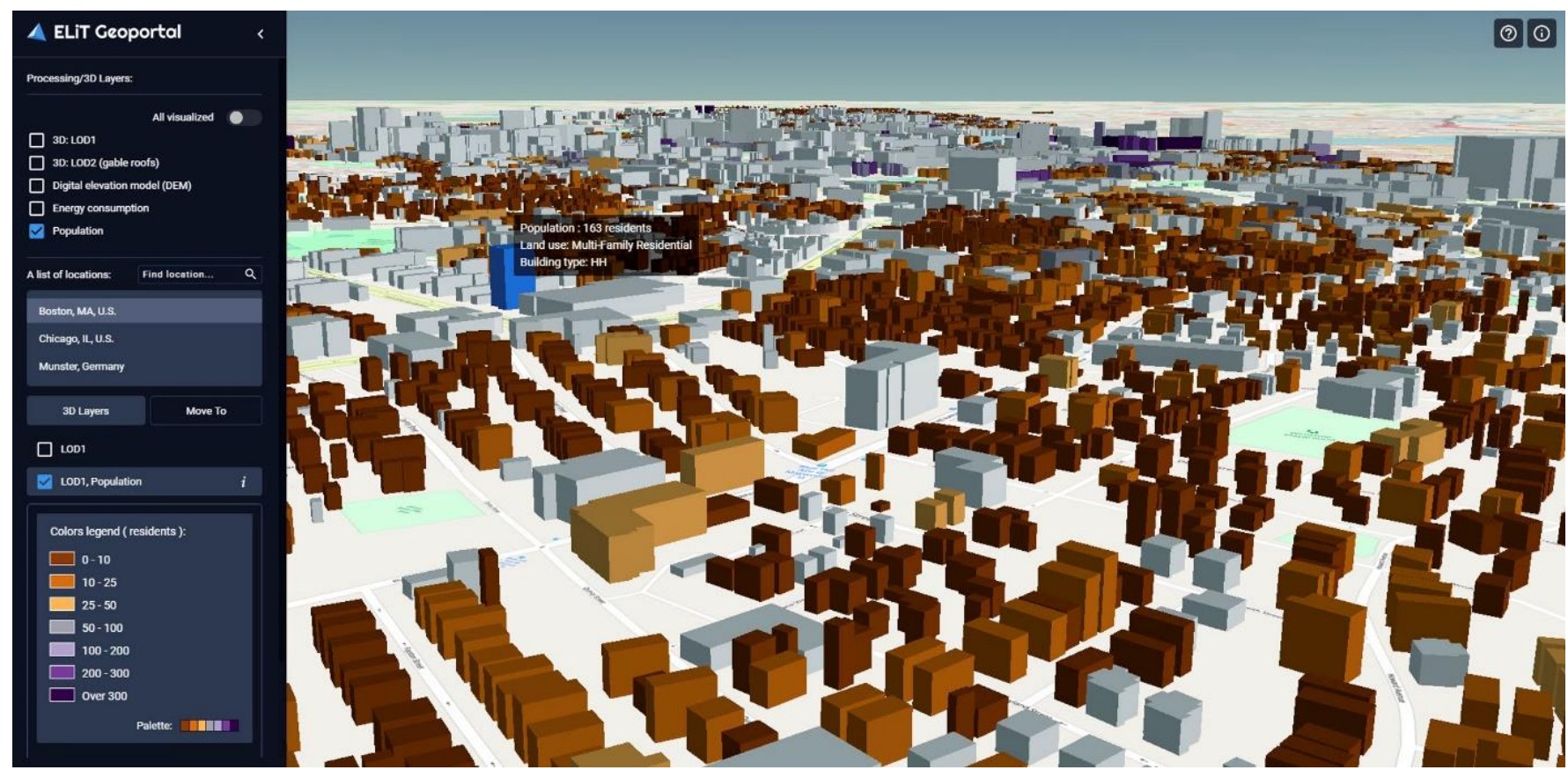

**Figure 9.** A fragment of a whole urban area of the city of Boston, Massachusetts, USA, presented on the Geoportal as 3D City GML LOD1 models with the implemented use case of population estimation with building geometries.

3.2.2. Use Case Energy

This applied use case has been implemented with similar to the UCP technological scheme, although evidently it has been based on other thematic basics [95–99]. Similar to the UCP workflow all processing is completed at the EGP backend and visualized then at the frontend with Cesium JS (Figure 10).

A use case aim is to define a value of average annual energy consumption (AAEC) of a building due to both heating in winter, and cooling in summer. The assignable for energy consumption area of a given building, a, has to be computer according to the European Union standards, and on the base of the technique from [96,97]. Thus, the finalizing characteristic value of building energy can be computed by dividing the actual energy consumption of a building by the determined assignable area—kWh/m$^2$a. For those EGP locations, where the UCE was implemented (Boston, Munster, and the city of Lyon, France), its workflow was accomplished in the similar way: 1. Sets of .OBJ files and .JSON files corresponding to 3D CityGML LOD1 models have been obtained for a given location by opensource Lidar data (both .LAZ, and .LAS formats) processing with *ELiT* software. 2. On the base of the UCP information sources, from which the urban land-use classes have been obtained, and according to the empirical content of some existing references besides those already mentioned above [100,101], we have made the following assumption. Upon a condition of all other equal factors non-residential (commercial) buildings may consume the energy up to 15–30% fewer, that residential ones. Thus, the key semantic attribute is Building Function Type. Due to the lack of semantic data we have implemented our original technique of automated definition of building type by its topology and geometry (ADBTG). For this purposes CityGML LOD2 models have been generated for those buildings in a given LOD 1 location, which type could not be defined by existing semantic data. 3. Another key attribute is constructionYear, according to an accepted regularity: the older a building, the more energy it consumes [98,100]. 4. One more attribute—storeysAboveGround is either obtained from the OSM-source, or calculated as the similar parameter, which is in the UCP. 5. If there is the information (from OSM or form some municipal sources) assignable for some separate buildings, which concerns values kWh/m$^2$a and kWh/m$^2$ year, it should be used for comparison of actual values versus calculated ones. 6. Standard types of building in the UCE have been defined by the *ELiT* software in the automated mode by the ADBTG technique, and this attributive information is prescribed to .OBJ/.JSON files (these acronyms correspond to their German equivalents [97,98]: detached single family house—EFH; row or twin house—RDH; small multifamily house—KMFN; big multifamily house—GMFN; multi-storey buildings—HH). One more key semantic attribute—BuildingType is generated and associated with all building models. 7. An assignable heating area and a heating volume are calculated for any building in a given territory (either a whole EGP location, or an AOI), which preferably should contain at least some actual values of actual energy consumption by buildings. 8. On the base of Steps 1–7 the semantic matrix of the building energy consumption has been built. This matrix contains the AAEC classes (A-G), which (1) are defined by a year of building construction, and (2) are divided for subclasses (e.g., subclass A-EFH) by the ADBTG building type (Step 6). 9. Each cell of this matrix obtain its default kWh/m$^2$ a value, in this way each cell of the matrix serves as a template for assigning the AAEC value for a certain building in an area, in a case of complete absence of any relevant semantic information. Thus, the UCE can be implemented either on the base of real attributive information, or by using this semantic matrix. 10. In both cases a given EGP Scene of the location is visualized in the standard EU legend of the building energy consumption.

In addition to the EGP location of Munster, the UCE has been provided for the city of Boston, Massachusetts, USA, and for Lyon, France. That we can see in the Geoportal interface (Figure 10), where A list of locations has been filtered by the Energy consumption checkbox turned on.

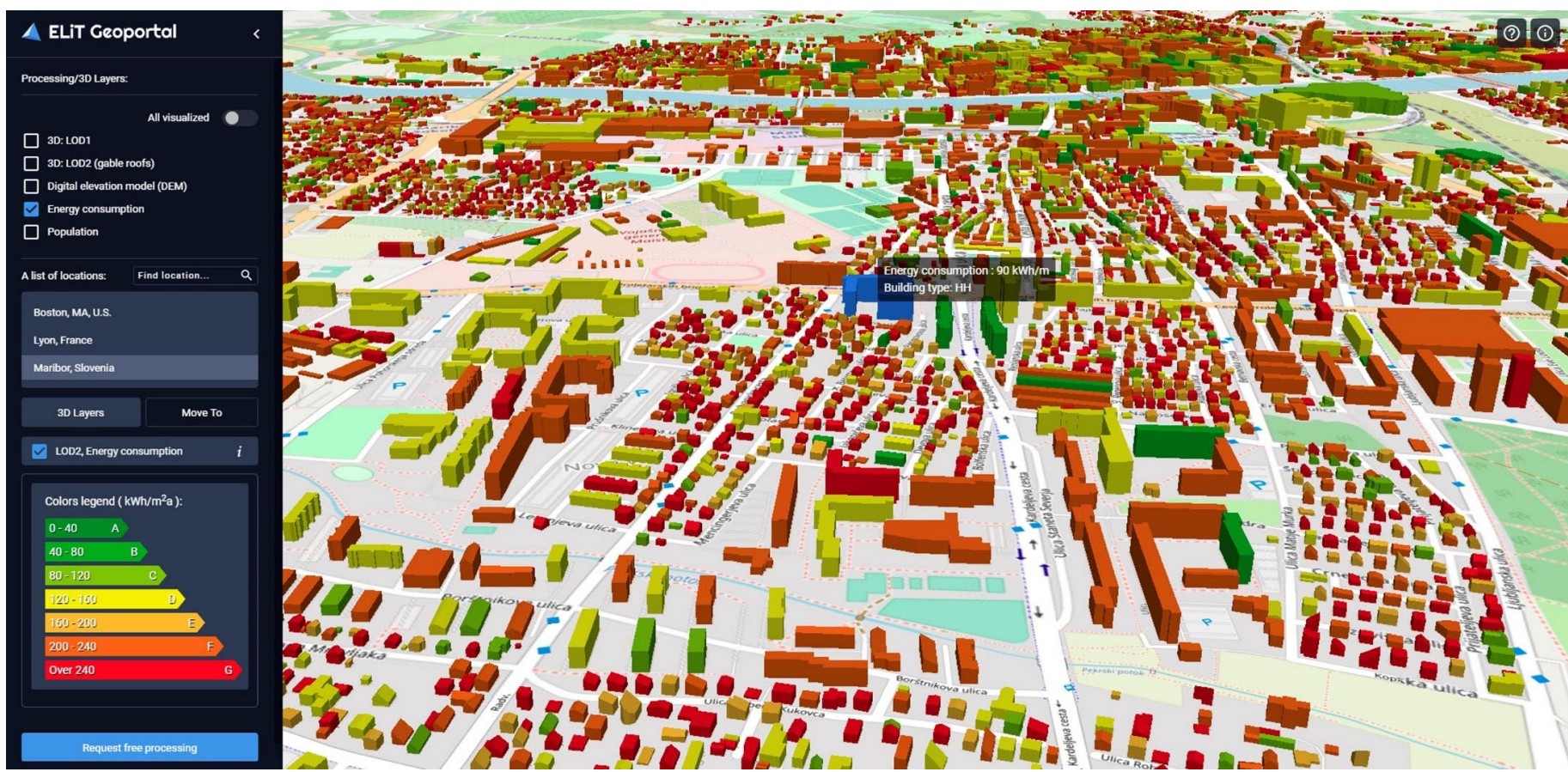

**Figure 10.** A fragment of a whole urban area of the city of Munster, Germany, presented on the Geoportal as 3D City GML LOD1 models with the implemented use case of estimation of energy consumption by buildings.

## 4. Discussion

The first topic of discussion may be the usage of either the high polyhedral modeling algorithms, including the classifying solution that has been examined in details in this text, or the low polyhedral ones already introduced in our previous publications [53,54]. On the base of a point altitude from an average DEM surface this classifying algorithm separates raw airborne LiDAR points for two categories. The first one contains the ground points, that form our original building footprint. The second category contains non-ground points, that are clustered with this building footprint. A relevant point cluster normally would represent a single building or a tree.

Contrary to this workflow, the low polyhedral modeling approach is strongly associated with "external footprints", what means they are obtained from the third party's sources, e.g., from the OSM resources. These footprints are employed to extract 3D point clouds on roofs. An obvious advantage of the HPM approach can be evidently seen then, because a lack of reliable footprint polygons is a well-known fact. In particular, these features globally cover substantially fewer areas, than LiDAR surveys do.

Another strong performance of the presented here combined (DEM-G + AFE) algorithmic solution is that efficient one with many refining steps (set interactively—Figure 2), which produces both continuous object (topography), and discrete ones (urban features) within a common algorithmic workflow, what understandably provides additional failure tolerance for both algorithmic branches (Figure 1). The presented two-branched classifying algorithm proves, that the HPM is mainly based on the point cloud classification, otherwise the LPM—on its largescale segmentation and further clustering into individual roof parcels. The examined classifying algorithm also provides some segmentation of a small scale, but it resulted in numerous polygonal segments that represented quite a rough surface of any building part while zoomed in (Figure 4). Contrary to it, the "LPM buildings" are the sets of planes with smoothed facets (Figure 5). In any case, a whole urban area may be a subject of differentiated application of some kind of "hybrid" HPM/LPM approach, when high-rise downtown buildings are extracted by the high polyhedral modeling (building extraction functionality), while low-rise ones of city outskirts and the nearest rural territories—by the LPM (building extraction rural area functionality).

The presented overall architectural scheme of the *ELiT* software family, which combines a desktop, a web-server, and a cloud application, may be accepted as an optimal solution for multifunctional Lidar data processing for the purposes of urban studies (Figure 3). Either processing and modeling with *ELiT*Core, or with *ELiT* Server, as well as with *ELiT* Geoportal (*ELiT* Cloud) can be defined as the most preferable one for resolving various use-cases in an urban block, a district, or in a whole city scope: applying of a particular software product in dependence of a defined modeling task.

Within the frameworks of a visualizing issue we have employed an open specification for visualizing massive 3D geospatial web-content—Cesium 3D Tiles in our cloud-based application. Rendering optimization with 3D Tiles has been implemented by comparison of two solutions that differ in their algorithmic contents titled in this text above as "(1)-Solution" versus "(2)-Solution", which has been presented in details. The latter, (2)-Solution, is our original one that can be parallelized, while (1)-Solution cannot. Moreover, there can be different counts of model numbers or a tile size in a case of (2)-Solution. Let us title these counts as "the initial 3D Tiles option" (for one feature model in a tile) versus "the advanced 3D Tiles option" (several feature models in a tile). It has been determined that the optimal structuring parameters for "the advanced option" is a tile edge of 512 m, and one tileset covers then up to 664 km$^2$ of urban area. Parametric characteristics of the EGP locations (Table 1) have served as a basis for a comparison of rendering for two options of our "(2)-Solution". Since we do not have an opportunity to provide more or less complete numeric analysis for such comparison in this concluding text, therefore we emphasize only its few representative terms:

1. Understandably, the latency time (a complete page loading) is the longest one for the biggest location—#14. It is up to 24 minutes with "the initial option", and we have to take into account,

that only LOD1 boxes could be constructed for this location due to extremely low average lidar point density for it (0.6 points per square meter). Thus, applying "the advanced option" we have accelerated rendering up to 10 min for all features of the location to be rendered.

2. It is noteworthy, that there is no a direct dependency between a number of models and the latency of a given location page. Nonetheless, if existing this dependency is more evident for LOD1 locations, than for LOD2 ones.

3. The best improvements in rendering (only those locations were estimated, which would possess some significant number of models, at least—several tens of thousands), while comparing "the initial option" versus "the advanced one", can be observed: for location #3 the latency time has been reduced from 14 min to 6 min; for location #4: from 17 to 8; location #15 is being rendered due to some reasons (probably, because of numerous LOD2 models present) even longer, than location #4, despite it has fourfold fewer models; latency time for this location with "the initial option" is 18 min, and it diminishes to 7 min with "the advanced option".

4. Other locations, which demonstrate 2.5–4 times speeding up in rendering, are ##1, 6, and 17 (Table 1).

5. Most of other locations present either none, or only slight speeding up in rendering for "the advanced one", while we compare two "3D Tiles options".

6. There are no evidences for any location, when "the initial 3D Tile option" would have had the faster rendering, than "the advanced 3D Tile option" does.

## 5. Conclusions

The applied use-case implementation has finalized our multifunctional workflow of LiDAR data processing, modeling, and analyzing for the Urban Studies domain. The elaborated and implemented techniques for both the UCP, and the UCE imply backend processing that concerns LOD1/LOD2 model generation, as well as processing from the side of frontend for visualizing thematic float data with the 3D Tiles structure in a CesiumJS scene with further gradient coloring of the attributive geospatial classes. Had we had statistically significant actual attributive information due to both use cases, such a solution would have been accomplished as a unique technique for applied usage by municipalities. Thus, this workflow of use cases is not conclusive, it still has to be validated, and we have to complete a more refined workflow in future research.

Thus, we have presented a comprehensive multifunctional approach to the urban topography and discrete features automated extraction on the base of Airborne LiDAR data processing. This research has demonstrated its applicability for different datasets, that represent heterogeneous urban configurations. Exactly this approach provides the implementation of our original, uniquely complete R&D cycle (from the urban terrain generation and feature extraction by raw LiDAR data processing till applied thematic use cases based on the models obtained):

- Raw LiDAR data initial preprocessing;
- Choosing an appropriate (due to the data nature and local urban configurations) solution—either low polyhedral modeling, or high polyhedral one;
- If the latter is selected, not third party's footprints are involved, but the original ones are extracted according to the basic HPM algorithm;
- Provision of the completely original two-branched DEM-G/AFE classifying algorithm, following by the urban topography generation and the feature extraction with customized setting up of processing in particular algorithmic blocks;
- Enhancement of the existing architectural scheme of software family, shifting emphasis from a desktop to its web- and cloud-components, what allows to process huge data volumes;
- Multifunctional application of software key functionalities: BE, BERA, CD, and DEM-G;
- Elaborating and establishing *ELiT* Geoportal as a cloud-based application within the frameworks of a service-oriented web-technology;

- Stuffing the Geoportal with the projects of 3D CityGML LOD1/LOD2 models;
- Accomplishing the original visualizing algorithmic solution, that consists of two options, based on optimizing the Cesium 3D Tiles structure for more efficient rendering of urban features on the Geoportal locations;
- Implementing practical thematic use cases for those locations, for which at least some semantic georeferenced data are available: Population Estimation with Building Geometries and Estimation of Energy Consumption by Buildings for Heating and Cooling;
- Upon these use cases' realization some supplementary unique solutions have been provided, e.g., our original technique of automated definition of building type by its topology and geometry.

The presented steps have pieced the complete R&D cycle of LiDAR data processing and obtaining derivative results outlined in this research.

**Author Contributions:** Conceptualization, Sergiy Kostrikov; methodology, Sergiy Kostrikov and Rostyslav Pudlo; software, Rostyslav Pudlo, Dmytro Bubnov, and Vladimir Vasiliev; validation, Rostyslav Pudlo, Dmytro Bubnov, and Vladimir Vasiliev; formal analysis, Sergiy Kostrikov, Rostyslav Pudlo, Dmytro Bubnov, and Vladimir Vasiliev; resources, Vladimir Vasiliev; data curation, Rostyslav Pudlo and Dmytro Bubnov; writing—original draft preparation, Sergiy Kostrikov and Rostyslav Pudlo; writing—review and editing, Sergiy Kostrikov, Rostyslav Pudlo, Dmytro Bubnov, and Vladimir Vasiliev; supervision, Sergiy Kostrikov; project administration, Vladimir Vasiliev. All authors have read and agreed to the published version of the manuscript.

**Funding:** This research received no external funding.

**Conflicts of Interest:** The authors declare no conflict of interest.

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
