# Peer review of "ELiT, Multifunctional Web-Software for Feature Extraction from 3D LiDAR Point Clouds"

_ijgi, doi:10.3390/ijgi9110650_

Round 1

Reviewer 1 Report

Thank you for addressing my suggestions, I appreciate the work done to improve the manuscript. I recommend it for further publication in IjGI.

Reviewer 2 Report

The authors have addressed the comments highlighted in the previous review process.

This manuscript is a resubmission of an earlier submission. The following is a list of the peer review reports and author responses from that submission.

Round 1

Reviewer 1 Report

From the article, it could be seen that the authors have collected or used sets of datasets using the proposed approach. However, in my opinion the paper has some shortcomings that might need to be improved. Some remarks that may possibly improve the quality of the paper are provided as follows:

  1. The title is confusing. Maybe it could be rephrased to reflects the main contribution of the article.
  2. It is a lengthy research article. I may suggest that this article to be shortened and make it focused on what is the main problem that the article trying to solve and prove (if accepted for publication/revision).
  3. Not a well-structured paper. The paper did not address the gaps as what have been done recently from the aspect of LiDAR technology, 3D city modeling, and features extraction (semi or automatic). I would suggest a specific section addressing the gaps would be appropriate.
  4. This article may have its own contribution. However, it is hard to perceive the significant contribution of the paper to the Geoinformation knowledge. The article could highlight the contribution from several aspects and justified with some references from recent literatures.
  5. Line 92-93: “LiDAR data have been recognized as the most preferable ones according to relatively low cost,….”, need a reference to this statement.
  6. The references used are a lot. However, most of the references cited in the article are not recent (<5 years). Around 17% from the 92 references are recent.

Author Response

Please see the file attachment.

Reviewer 2 Report

The review of ijgi-921406 (round 1)

The subject undertaken in the reviewed manuscript (web-based AFE software) is valuable and interesting for the audience of IjGI. However, the confused manuscript structure, makes the manuscript substantive evaluating difficult.

The introduction (section 1, L32-148) can be accepted, but the next half of the section (L149 ~300), declared as the method (L149), are a kind of literature review, not authors original methodology. This error is being systematically repeated e.g. 485-508, e.g. L725-729 ( L725 and the following belongs to the method, not a result). Therefore I do suggest to select only the most important pieces of information and the most relevant references to write up to ~3 pages of essential introduction, subsequently divided into thematic subsections. 

Furthermore, if something seems to be a quite clear – as Authors has stated e.g. In L509, therefore it is not worth to be mentioned in the scientific article. A statement like (…) Without any additional reference we reiterate a commonly known fact, (…) (e.g. L 510) disqualify the manuscript as a scientific one.

After refining the introduction, please move on to the method (like it is done in L348; L339-347 are kind of discussion, not a method description). To strength the manuscript structure, I would suggest using numbered stages for the method description if possible.

Also, figures need to be explained more carefully. Currently, L 348 says (…) According to the flowchart of the algorithm shown in Figure 1 the step-by-step description of its Ground branch (the second, right, algorithmic brunch) may (…)

Truly, I haven’t found Figure 1 in the reviewed manuscript, only Figure 1a; 1b. Probably they should be numbered as Figure 1 and Figure 2. If possible, use a single figure (e.g. Fg. 1) to explain the algorithm flowchart and if necessary use Fig. 1a, b, c to provide some details.

The same referees to “blocks”. For example, the “block 2.3” is mentioned in L367 & L452, however, I have found it difficult to find the block number (any blocks) on the presented flowchart. Different geometric figures are being used for flowchart presentation – do they have any meaning, are they necessary? Maybe they can be uniformed and numbered according to block number. If yes, please provide sufficient description and blocks numbers. Please also pay more attention to the figure legibility (the white background suggest that graphic layout should be performed more effectively). Please look through the manuscript and consider its structure – for example Fig. 7 - shouldn't belong to the method section?

Finally, the tithe should be simplified, please consider the phrase “ELiT, cloud-based software for 3D point cloud feature extraction”.

Please also explain “the urban DEM” – how is it different from the Digital Elevation Model and Digital Surface Model (DSM). I guess the authors are thinking about the second one, but I do believe that “urban” is unnecessary. Also consider using “3D point cloud” instead of LiDAR, because the 3D point cloud is being used for feature extraction, not LiDAR. This applies to the whole manuscript, except L152-153.

Concluding, the main drawback of the manuscript is wrong structure, which I do believe can be improved to enable substantive manuscript evaluation in the next review round. I encourage the Authors to make the manuscript more concise (reduce the number of pages) to underline the valuable research outcomes.

Review date: 31 August 2020

Author Response

Please see the file attachment
